# Training-Free Structured Diffusion Guidance for Compositional Text-to-Image Synthesis

**Weixi Feng[1], Xuehai He[2], Tsu-jui Fu[1], Varun Jampani[3], Arjun Akula[3],**
**Pradyumna Narayana[3], Sugato Basu[3], Xin Eric Wang[2], William Yang Wang[1]**
[1]University of California, Santa Barbara, [2]University of California, Santa Cruz, [3]Google

## Abstract

Large-scale diffusion models have achieved state-of-the-art results on text-to-image synthesis (T2I) tasks. Despite their ability to generate high-quality yet creative images, we observe that attribution-binding and compositional capabilities are still considered major challenging issues, especially when involving multiple objects. Attribute-binding requires the model to associate objects with the correct attribute descriptions, and compositional skills require the model to combine and generate multiple concepts into a single image. In this work, we improve these two aspects of T2I models to achieve more accurate image compositions. To do this, we incorporate linguistic structures with the diffusion guidance process based on the controllable properties of manipulating cross-attention layers in diffusion-based T2I models. We observe that keys and values in cross-attention layers have strong semantic meanings associated with object layouts and content. Therefore, by manipulating the cross-attention representations based on linguistic insights, we can better preserve the compositional semantics in the generated image. Built upon Stable Diffusion, a SOTA T2I model, our structured cross-attention design is efficient that requires no additional training samples. We achieve better compositional skills in qualitative and quantitative results, leading to a significant 5-8% advantage in head-to-head user comparison studies. Lastly, we conduct an in-depth analysis to reveal potential causes of incorrect image compositions and justify the properties of cross-attention layers in the generation process.

## 1 Introduction

Text-to-Image Synthesis (T2I) is to generate natural and faithful images given a text prompt as input. Recently, there has been a significant advancement in the quality of generated images by extremely large-scale vision-language models, such as DALL-E 2 (Ramesh et al., 2022), Imagen (Saharia et al., 2022), and Parti (Yu et al., 2022). In particular, Stable Diffusion (Rombach et al., 2022) is the state-of-the-art open-source implementation showing superior evaluation metric gains after training over billions of text-image pairs.

In addition to generating high-fidelity images, the ability to compose multiple objects into a coherent scene is also essential. Given a text prompt from the user end, T2I models need to generate an image that contains all necessary visual concepts as mentioned in the text. Achieving such ability requires the model to understand both the full prompt and individual linguistic concepts from the prompt. As a result, the model should be able to combine multiple concepts and generate novel objects that have never been included in the training data. In this work, we mainly focus on improving the compositionality of the generation process, as it is essential to achieve controllable and generalized text-to-image synthesis with multiple objects in a complex scene.

*Attribute binding* is a critical compositionality challenge (Ramesh et al., 2022; Saharia et al., 2022) to existing large-scale diffusion-based models. Despite the improvements in generating multiple objects in the same scene, existing models still fail when given a prompt such as "a brown bench in front of a white building" (see Fig. 1). The output images contains "a white bench" and "a brown building" instead, potentially due to strong training set bias or imprecise language understanding. From a practical perspective, explaining and solving such a two-object binding challenge is a primary step to understanding more complex prompts with multiple objects. Therefore, how to bind the attributes

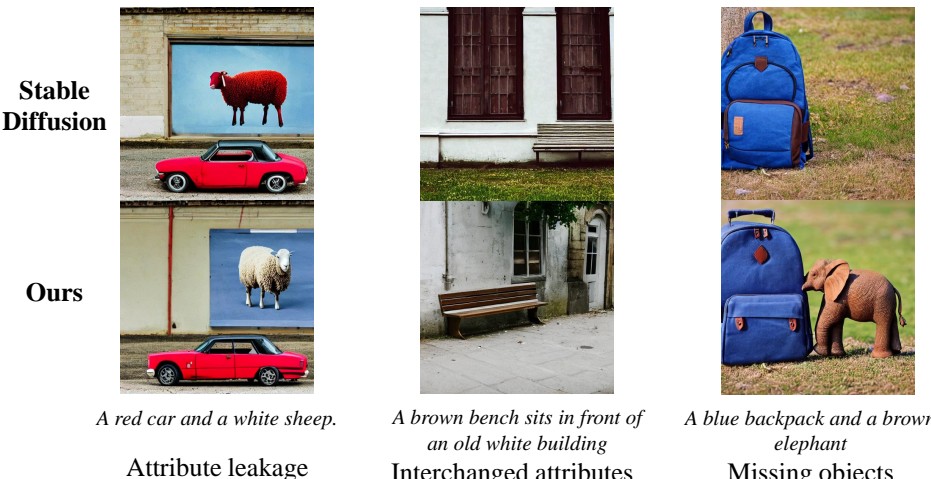



Stable Diffusion

Ours

*A red car and a white sheep.*

Attribute leakage

*A brown bench sits in front of an old white building*

Interchanged attributes

*A blue backpack and a brown elephant*

Missing objects



Figure 1: **Three challenging phenomena in the compositional generation.** Attribute leakage: The attribute of one object is (partially) observable in another object. Interchanged attributes: the attributes of two or more objects are interchanged. Missing objects: one or more objects are missing. With slight abuse of attribute binding definitions, we aim to address all three problems in this work.

to the correct objects is a fundamental problem for a more complicated and reliable compositional generation. While previous work has addressed compositional T2I (Park et al., 2021), our work tackles open-domain foreground objects with counterfactual attributes, such as color and materials.

Even though state-of-the-art (SOTA) T2I models are trained on large-scale text-image datasets, they can still suffer from inaccurate results for simple prompts similar to the example above. Hence, we are motivated to seek an alternative, data-efficient method to improve the compositionality. We observe that the attribute-object relation pairs can be obtained as text spans for free from the parsing tree of the sentence. Therefore, we propose to combine the structured representations of prompts, such as a constituency tree or a scene graph, with the diffusion guidance process. Text spans only depict limited regions of the whole image. Conventionally, we need spatial information such as coordinates (Yang et al., 2022) as input to map their semantics into corresponding images. However, coordinate inputs cannot be interpreted by T2I models. Instead, we make use of the observations that attention maps provide free *token-region associations* in trained T2I models (Hertz et al., 2022). By modifying the key-value pairs in cross-attention layers, we manage to map the encoding of each text span into attended regions in 2D image space.

In this work, we discover similar observations in Stable Diffusion (Rombach et al., 2022) and utilize the property to build structured cross-attention guidance. Specifically, we use language parsers to obtain hierarchical structures from the prompts. We extract text spans across all levels, including visual concepts or entities, and encode them separately to disentangle the attribute-object pairs from each other. Compared to using a single sequence of text embedding for guidance, we improve the compositionality by multiple sequences where each emphasizes an entity or a union of entities from multiple hierarchies in the structured language representations. We refer to our method as Structured Diffusion Guidance (StructureDiffusion). Our contributions can be summarized as three-fold:

- We propose an intuitive and effective method to improve compositional text-to-image synthesis by utilizing structured representations of language inputs. Our method is efficient and training-free that requires no additional training samples.

- Experimental results show that our method achieves more accurate attribute binding and compositionality in the generated images. We also propose a benchmark named **A**ttribute **B**inding **C**ontrast set (ABC-6K) to measure the compositional skills of T2I models.

- We conduct extensive experiments and analysis to identify the causes of incorrect attribute binding, which points out future directions in improving the faithfulness and compositionality of text-to-image synthesis.

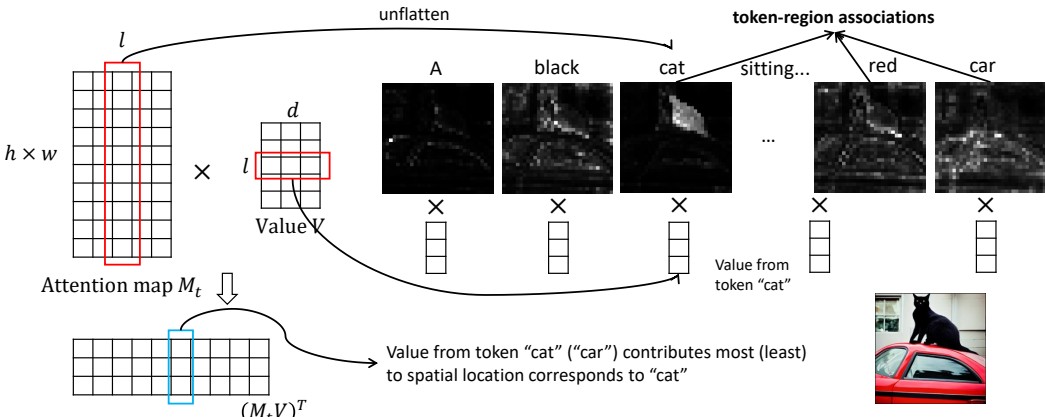

Figure 2: An illustration of cross-attention operations and the token-region associations from attention maps. We omit some tokens for simplicity.

## 2 DIFFUSION MODELS & STRUCTURED GUIDANCE

In this section, we propose a simple yet effective approach incorporating structured language representations into the cross-attention layers. We briefly introduce the Stable Diffusion model and its critical components in Sec. 2.1. Then, we present our method in detail in Sec. 2.2.

### 2.1 BACKGROUND

**Stable Diffusion** We implement our approach and experiments on the state-of-the-art T2I model, Stable Diffusion (Rombach et al., 2022). It is a two-stage method that consists of an autoencoder and a diffusion model. The pre-trained autoencoder encodes images as lower-resolution latent maps for diffusion training. During inference, it decodes generated outputs from the diffusion model into images. The diffusion model generates lower-resolution latent maps based on a random Gaussian noise input $z^T$. Given $z^T$, it outputs a noise estimation $\epsilon$ at each step $t$ and subtracts it from $z^t$. The final noise-free latent map prediction $z^0$ is fed into the autoencoder to generate images. Stable Diffusion adopts a modified UNet (Ronneberger et al., 2015) for noise estimation and a frozen *CLIP text encoder* (Radford et al., 2021) to encode text inputs as embedding sequences. The interactions between the image space and the textual embeddings are achieved through multiple *cross-attention layers* in both downsampling and upsampling blocks.

**CLIP Text Encoder** Given an input prompt $\mathcal{P}$, the CLIP encoder encodes it as a sequence of embeddings $\mathcal{W}_p = \text{CLIP}_{\text{text}}(\mathcal{P})$ where $c_p$ is the embedding dimension and $l$ is the sequence length. Our key observation is that the contextualization of CLIP embeddings is a potential cause of incorrect attribute binding. Due to the causal attention masks, tokens in the later part of a sequence are blended with the token semantics before them. For example, When the user indicates some rare color for the second object (e.g. "a yellow apple and red bananas"), Stable Diffusion tends to generate "banana" in "yellow", as the embeddings of "yellow" is attended by token "banana".

**Cross Attention Layers** The cross-attention layers take the embedding sequences from the CLIP text encoder and fuse them with latent feature maps to achieve classifier-free guidance. Denote a 2D feature map $\mathcal{X}^t$, it is projected into queries by a linear layer $f_Q(\cdot)$ and reshaped as $Q^t \in R^{(n,h \times w,d)}$ where $n$ denotes the number of attention heads, $d$ is the feature dimension. Similarly $\mathcal{W}_p$ is projected as keys and values $K_p, V_p \in R^{(n,l,d)}$ by linear layers $f_K(\cdot), f_V(\cdot)$. The attention maps refer to the product between queries and keys, denoted as a function $f_M(\cdot)$

$$M^t = f_M(Q^t, K_p) = \text{Softmax}(\frac{Q^t K_p^T}{\sqrt{d}}), \; M^t \in R^{(n,h \times w,l)}. \tag{1}$$

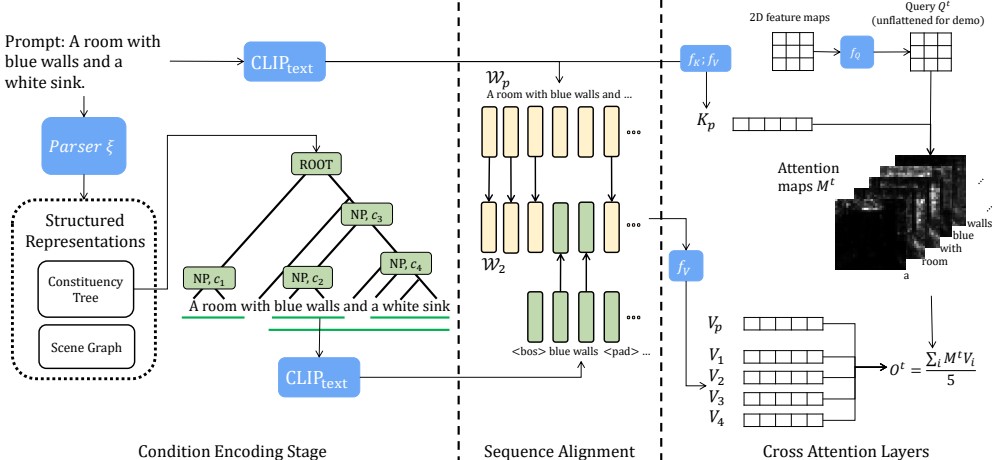

Figure 3: An illustration of our cross-attention design with structured representations. We unflatten the query and attention maps and omit the feature dimension $d$ of all query, key, and value tensors for demonstration purposes. Note that noun phrases at multiple hierarchies are extracted and encoded through the frozen CLIP text encoder and projected to value vectors.

**Cross Attention Controls**   Hertz et al. (2022) observes that the spatial layouts depend on the cross attention maps in Imagen Saharia et al. (2022). These maps control the layout and structure of generated images, while the values contain rich semantics mapped into attended regions. Therefore, we assume that the image layout and content can be disentangled by controlling attention maps and values separately.

## 2.2 STRUCTURED DIFFUSION GUIDANCE

Given the challenging prompts in Fig. 1, the attribute-object pairs are available **for free**[1] in many structured representations, such as a constituency tree or a scene graph. We seek an implicit way of combining language structures with the cross-attention layers. As is shown in Fig. 3, we can extract multiple noun phrases (NPs) and map their semantics into corresponding regions. Since $M_t$ provides natural token-region associations (see Fig. 2), we can apply it to multiple values from different NPs to achieve region-wise semantic guidance.

Specifically, given a parser $\xi(\cdot)$, we first extract a collection of concepts from all hierarchical levels as $\mathcal{C} = \{c_1, c_2, \ldots, c_k\}$. For constituency parsing, we extract all NPs from the tree structure (see Fig.3 left). For the scene graphs, we extract objects and their relations with another object as text segments. We encode each NP separately:

$$\mathbb{W} = [\mathcal{W}_{\mathrm{p}}, \mathcal{W}_1, \mathcal{W}_2, \ldots, \mathcal{W}_k], \ \mathcal{W}_i = \mathrm{CLIP}_{\text{text}}(c_i), \ i = 1, \ldots k. \tag{2}$$

The embedding sequence $\mathcal{W}_i$ is realigned with $\mathcal{W}_p$ as shown in the middle of Fig. 3. Embeddings between $\langle\mathrm{bos}\rangle$ and $\langle\mathrm{pad}\rangle$ are inserted into $\mathcal{W}_p$ to create a new sequence, denoted as $\overline{\mathcal{W}}_i$. We use $\overline{\mathcal{W}}_{\mathrm{p}}$ to obtain $K_p$ and $M^t$ as in Eq. 1, assuming that the full-prompt key is able to generate layouts without missing objects. We obtain a set of values from $\mathbb{W}$ and multiply each with $M^t$ to achieve a conjunction of $k$ NPs in $\mathcal{C}$:

$$\mathbb{V} = [f_V(\mathcal{W}_{\mathrm{p}}), f_V(\overline{\mathcal{W}}_1), \ldots, f_V(\overline{\mathcal{W}}_k)] = [V_{\mathrm{p}}, V_1, \ldots, V_k]. \tag{3}$$

$$O^t = \frac{1}{(k+1)} \sum_i (M^t V_i), i = \mathrm{p}, 1, 2, \ldots, k. \tag{4}$$

Compared to using $f_V(\mathcal{W}_p)$ only, Eq. 4 does not modify the image layout or composition since $M^t$ is still calculated from $Q^t, K_p$. Empirically, we justify the claim by a series of visualizations of $M_t$

---

[1]For free means that the extra computational cost introduced here is trivial in the whole diffusion process.

---

**Algorithm 1** StructureDiffusion Guidance.

---

**Require:**
    **Input:** Prompt $\mathcal{P}$, Parser $\xi$, decoder $\psi$, trained diffusion model $\phi$.
    **Output:** Generated image $x$.
1: Retrieve concept set $\mathcal{C} = [c_1, \ldots, c_k]$ by traversing $\xi(\mathcal{P})$;
2: $\mathcal{W}_p \leftarrow \text{CLIP}_{\text{text}}(\mathcal{P})$, $\mathcal{W}_i \leftarrow \text{CLIP}_{\text{text}}(c_i)$;                              $i = 1, \ldots, k$
3: **for** $t = T, T-1, \ldots, 1$ **do**
4:     **for** each cross attention layer in $\phi$ **do**
5:         Obtain previous layer's output $\mathcal{X}^t$.
6:         $Q^t \leftarrow f_Q(\mathcal{X}^t)$, $K_p \leftarrow f_K(\mathcal{W}_p)$, $V_i \leftarrow f_V(\overline{\mathcal{W}_i})$;         $i = p, 1, \ldots, k$
7:         Obtain attention maps $M^t$ from $Q^t, K_p$;                    {Eq. 1}
8:         Obtain $O^t$ from $M^t, \{V_i\}$, and feed to following layers;     {Eq. 4}
9:     **end for**
10: **end for**
11: Feed $z^0$ to decoder $\psi(\cdot)$ to generate x.

---

(see Appendix C). However, Stable Diffusion tends to omit objects in generated images (Fig. 1), especially for concept conjunctions that connect two objects with the word "and". We devise a variant of our method that computes a set of attention maps $\mathbb{M} = \{M_p^t, M_1^t, \ldots\}$ from $\mathcal{C}$ and multiply them to $\mathbb{V}$:

$$\mathbb{K} = \{f_K(\mathcal{W}_i)\}, \ \mathbb{M}^t = \{f_M(Q^t, K_i)\}, \ i = p, 1, 2, \ldots, k. \tag{5}$$

$$O^t = \frac{1}{(k+1)} \sum_i (M_i^t V_k), i = p, 1, 2, \ldots, k. \tag{6}$$

$O^t$ is the output of a certain cross-attention layer and the input into downstream layers to generate final image $x$. Our algorithm can be summarized as 1, which requires no training or additional data.

## 3 EXPERIMENT

### 3.1 EXPERIMENT SETTINGS

**Datasets** To address attribute binding and compositional generation, we propose a new benchmark, **A**ttribute **B**inding **C**ontrast set (ABC-6K). It consists of natural prompts from MSCOCO where each contains at least two color words modifying different objects. We also switch the position of two color words to create a contrast caption (Gardner et al., 2020). We end up with 6.4K captions or 3.2K contrastive pairs. In addition to natural compositional prompts, we challenge our method with less detailed prompts that conjunct two concepts together. These prompts follow the sentence pattern of "a red apple and a yellow banana" and conjunct two objects with their attribute descriptions. We refer to this set of prompts as **C**oncept **C**onjunction 500 (CC-500). We also evaluate our method on 10K randomly sampled captions from MSCOCO (Lin et al., 2014). We show that our method generalizes beyond attribute binding and introduces no quality degradation for general prompts.

**Evaluation Metrics** We mainly rely on human evaluations for compositional prompts and concept conjunction (ABC-6K & CC-500). We ask annotators to compare two generated images, from Stable Diffusion and our method respectively, and indicate which image demonstrates better image-text alignment or image fidelity. For image fidelity, we ask the annotators "Regardless of the text, which image is more realistic and natural?". We also investigate an automatic evaluation metric for image compositions, i.e., using a SOTA phrase grounding model GLIP (Li et al., 2022) to match phrase-object pairs. As for system-level evaluation, we follow previous work to utilize Inception Score (IS) (Salimans et al., 2016), Frećhet Inception Distance (FID) (Heusel et al., 2017) and CLIP R-precision (R-prec.) (Park et al., 2021). IS and FID mainly measure the image bank's systematic quality and diversity, while R-prec measures image-level alignment.

### 3.2 COMPOSITIONAL PROMPTS

Here we show the quantitative and qualitative evaluation results on ABC-6K. We observe that our method sometimes generates very similar images to Stable Diffusion. Hence, we first generate

| Benchmark | StructureDiffusion (ours) v.s. | Alignment | | | Fidelity | | |
|---|---|---|---|---|---|---|---|
| | | Win (↑) | Lose (↓) | Tie | Win (↑) | Lose (↓) | Tie |
| **ABC-6K** | Stable Diffusion | **42.2** | 35.6 | 22.2 | **48.3** | 39.1 | 12.6 |
| **CC-500** | Stable Diffusion | **31.8** | 27.7 | 38.9 | **37.8** | 30.6 | 31.6 |
| | Composable Diffusion | **46.5** | 30.1 | 22.8 | **61.4** | 19.8 | 18.8 |

Table 1: Percentage of generated images of StructureDiffusion that are better than (win), tied with, or worse than (lose) the compared model in terms of text-image alignment and image fidelity. We filtered out 20% most similar image pairs for comparison (See Sec. E). Composable Diffusion cannot be applied to ABC-6K as those prompts may not contain explicit "and" words that separate concepts.

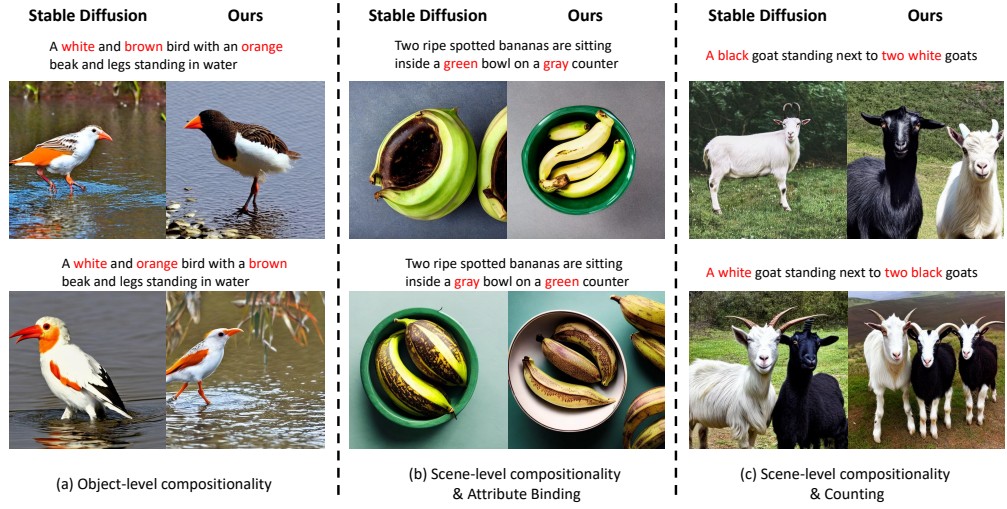

(a) Object-level compositionality

(b) Scene-level compositionality & Attribute Binding

(c) Scene-level compositionality & Counting

Figure 4: Qualitative results on ABC-6K. Our method improves both object-level and scene-level compositionality.

two images per prompt for our method and Stable Diffusion, involving around 12K image pairs to compare. Then, we filter out 20% of the most similar pairs and then randomly sampled 1500 pairs for human evaluations. As shown in Table 1, annotators indicate around a 42% chance of our method winning the comparison, 7% higher than losing the comparison. There is still a 22% of chance that our images are tied with images from Stable Diffusion.

We show qualitative examples characterizing three different perspectives in Fig. 4. Our method fills in the correct color for different parts of an object or different objects, as shown in the first two examples. The third example demonstrates that our method can mitigate the issue of "missing objects". Among the 42% winning cases, there are 31% for "fewer missing objects", 14.1% for "better-matched colors", and 54.8% for "other attributes or details" as indicated by annotators. The results certify that the improvement goes beyond colors to component completeness and fine-grained details. More qualitative examples characterizing all three aspects can be found in Fig. 14 in the Appendix.

## 3.3 CONCEPT CONJUNCTION

Here we address challenging concept conjunction prompts and evaluate our method on CC-500. Apart from Stable Diffusion, we also compare to Composable Diffusion (Liu et al., 2022) implemented on top of Stable Diffusion. For Composable Diffusion, we separate the prompts into text segments by the keyword "and" and feed each span into an independent diffusion process. We generate three images per prompt and use all images for human evaluation for Stable Diffusion. We randomly sampled 600 images for comparison to Composable Diffusion.

| Methods | CC-500 (Prompt format: "a [colorA] [objectA] and a [colorB] [objectB]" ) | | | | | |
| | Human Annotations | | | GLIP | | |
| | Zero/One obj. ($\downarrow$) | Two obj. | Two obj. w/ correct colors | Zero/One obj. ($\downarrow$) | Two obj. | Human-GLIP Consistency |
| --- | --- | --- | --- | --- | --- | --- |
| **Stable Diffusion** | 65.5 | 34.5 | 19.2 | 69.0 | 31.0 | 46.4 |
| **Composable Diffusion** | 69.7 | 30.3 | 20.6 | 74.2 | 25.8 | 48.9 |
| **StructureDiffusion (Ours)** | **62.0** | **38.0** | **22.7** | **68.8** | **31.2** | 47.6 |

Table 2: Fine-grained human and automatic evaluation results on CC-500. Recall that each prompt is a conjunction of two different objects with different colors. "Zero/One obj." means that the model fails to generate all desired objects in the image. "Human-GLIP consistency" reflects the percentage of images where human annotations align with GLIP detection results.

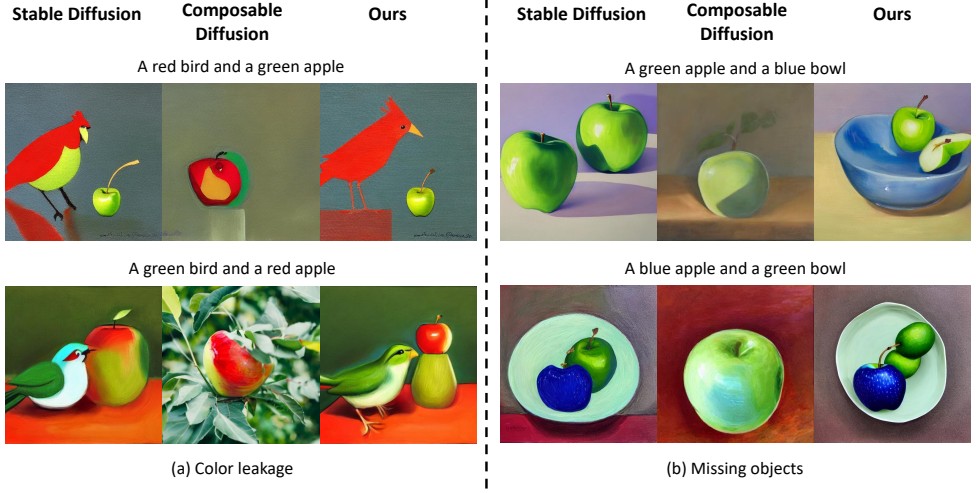

Figure 5: Qualitative results on CC-500 prompts that emphasize two aspects. (a) Color leakage: our method prevents the green color from invading the bird or apple. (b) Missing objects: our method completes the "blue bowl" and improves the quality of the "blue apple".

As shown in Table 1, our method outperforms Stable Diffusion by around 4.1% and Composable Diffusion by 16.4% in terms of image-text alignment. We also observe that our method enhances some fine-grained details in the generated images, leading to a 7.2% improvement in image fidelity when compared with Stable Diffusion. We observe that images from composable diffusion can be oversaturated with unnatural visual textures and layouts, which could be the reason for StructureDiffusion to have high win rate in image fidelity. As shown in Fig. 5 and Fig. 13. Our approach prevents color bleeding (left), missing objects (right) and strengthens details (right).

To further quantify the text-image alignment, we consider both human annotations and automatic evaluations. For each object mentioned in the prompt, we ask annotators whether the object exists in the image and whether it is in the correct color. We also apply a state-of-the-art detection model GLIP (Li et al., 2022) to ground each "a [color] [object]" phrase into bounding boxes. We report the percentage of images that contain incomplete objects / complete objects / complete objects with correct colors in Table 2. StructureDiffusion improves the compositionality by 3.5% based on human annotations while only 0.2% based on GLIP. We discover that humans disagree with GLIP for more than 50% of the images, as entailed by the low consistency rate. Previous work also suggests the deficiency of large pre-trained models in compositional understanding (Thrush et al., 2022).

### 3.4 OTHER PROMPTS

We show that our StructureDiffusion maintain the overall image quality and diversity on general prompts. We follow the standard evaluation process and generate 10,000 images from randomly

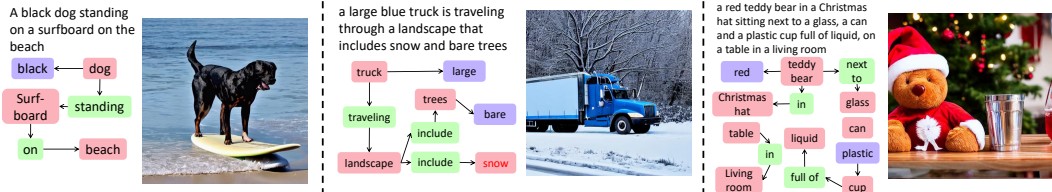

Figure 6: Qualitative results of using scene graph parser to generate structured representations.

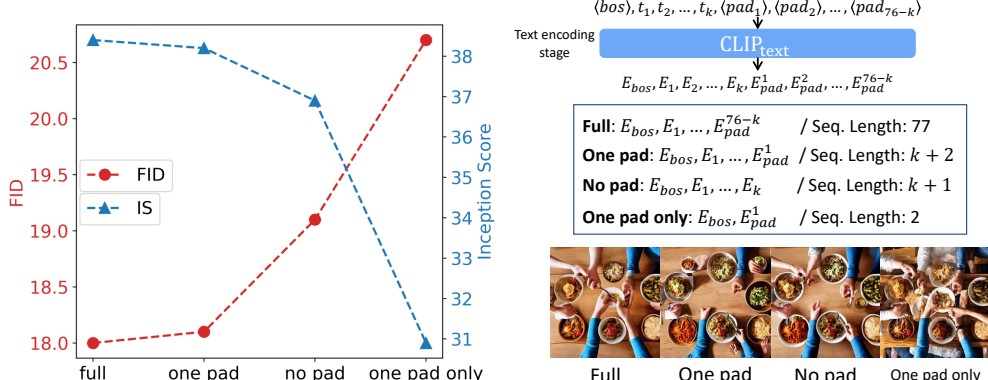

Figure 7: Ablation study on the text sequence embeddings. We find that the padding embeddings are fully contextualized, representing the prompt's high-level semantics. However, not all padding tokens are necessary to maintain a high-fidelity output from Stable Diffusion.

sampled MSCOCO captions. Stable Diffusion obtains 39.9 IS, 18.0 FID and 72.2 R-Precision. Our method achieves 40.9 IS, 17.9 FID and 72.3 R-Precision. StructureDiffusion maintains the image fidelity and diversity as indicated in the comparable IS/FID/R-Prec scores.

## 3.5 SCENE GRAPH INPUT

We show that our method is not limited to constituency parsing but can also be extended to other structured representations, such as scene graphs. As shown in Fig. 6, we first adopt the scene graph parser (Wu et al., 2019) and obtain a graph like the ones next to each image from the input prompt. The parser returns basic entities and their relations in between. We extract text spans of basic entities with their attributes attached and text spans that include two related entities. We provide examples in Appendix 3 and make comparison to the constituency parser. Similarly, we encode these spans separately and re-align each with the entire prompt encoding sequence. On MS-COCO, the scene graph parser setting maintains the image quality with 39.2 IS, 17.9 FID, and 72.0 R-Precision. When compared to Stable Diffusion on ABC-6K, the scene graph parser achieves 34.2%-32.9%-32.9% Win-Lose-Tie in image-text alignment and 34.5%-32.5%-33.0% Win-Lose-Tie in image fidelity. As for CC-500, the scene graph parser leads to the same output images due to the same text spans. We refer to Table 3 and Fig. 12 for more results and comparison.

## 4 ABLATION STUDY

### 4.1 RE-ALIGNING SEQUENCE

In Section 2, we describe a method to realign the encoding of a text span back into the sequence of the full prompt. Since the noun-phrase text spans are shorter than the full sequence, re-alignment ensures that each token's value vector corresponds to the correct attention map. On the other hand, naively expanding the span to the length of the full sequence degrades the image quality by $\sim$2 IS / FID (37.5 IS, 19.8 FID) compared to images with re-alignment or Stable Diffusion.

## 4.2 Contextualized Text Embeddings

One limitation brought by our StructureDiffusion is that the cross-attention computation costs increase by the number of noun phrases. Yet we noticed that most of the attention maps are computed from padding embeddings, as Stable Diffusion adopts CLIP text encoders and automatically pads the sequence to 77 tokens. We conjecture that not all padding tokens are necessary for generating high-quality images. As is shown in Fig. 7, we study four different patterns of token embeddings. We discover that leaving the nearest padding embeddings maintains a similar IS / FID score as the full sequence. Further removing this padding embedding results in apparent degradation. While only using the nearest padding embedding results in the worst image quality, we find that the high-level image layout and semantics are preserved (see bottom right of Fig. 7). This phenomenon indicates that the padding embeddings are fully contextualized with the full prompt semantics. This also justifies our re-alignment operation that preserves padding embeddings of the main sequence $\mathcal{W}_{\text{full}}$.

## 5 Related Work

**Text-to-Image Synthesis** The diffusion model is an emerging type of model that generate high-quality images with a much more stable training process (Song & Ermon, 2019; Ho et al., 2020). Rombach et al. (2022) proposes to encode an image with an autoencoder and then leverage a diffusion model to generate continuous feature maps in the latent space. Stable Diffusion Rombach et al. (2022) adopts similar architecture but is trained on large-scale image-text datasets with fixed CLIP text encoder. Imagen (Saharia et al., 2022) addresses the importance of language understanding by using a frozen T5 encoder (Raffel et al., 2020), a dedicated large language model. We mainly focus on diffusion models and conduct our experiments on Stable Diffusion (Rombach et al., 2022), the SOTA open-sourced T2I model.

**Compositional Generation** The compositional or controllable generation has been an essential direction for T2I models to understand and disentangle basic concepts in the generation process. As text inputs are relatively weak conditions, previous work leverage layout or scene graph to enhance compositionality (Johnson et al., 2018; Hong et al., 2018; Yang et al., 2022; Gafni et al., 2022). More recently, Liu et al. (2022) proposes an approach where the concept conjunctions are achieved by adding estimated scores from a parallel set of diffusion processes. In contrast, our method can be directly merged into the cross-attention layers with much less computational overhead.

**Diffusion Guidance** Ho & Salimans (2022) develops classifier-free guidance where a single diffusion model is jointly trained under conditional and unconditional inputs. Most large-scale SOTA models, including autoregressive ones, adopt this technique for flexible and improved conditional synthesis results (Rombach et al., 2022; Ramesh et al., 2022; Gafni et al., 2022; Yu et al., 2022; Saharia et al., 2022). Hertz et al. (2022) discovers unique properties of cross attention maps on Imagen (Saharia et al., 2022) and achieves structure-preserving image editing by manipulating these maps. We observe similar properties in Stable Diffusion (Rombach et al., 2022) but propose a different algorithm for fine-grained, compositional text-to-image generation.

## 6 Conclusion

In this work, we propose a training-free method for compositional text-to-image generation. First, we observe that existing large-scale T2I diffusion models can still struggle in compositional image synthesis. We address this challenge by explicitly focusing on binding objects with the correct attributes. Second, we propose structured diffusion guidance incorporating language structures into the cross-attention layers. We propose two simple techniques to align the structured encoding with the attention maps. Using our structured guidance on Stable Diffusion, attributes can be bound more accurately while maintaining the overall image quality and diversity. In addition, we justify our approach by conducting an in-depth analysis of the frozen language encoder and attention maps. Future work may explore explicit approaches to generate plausible image layouts without missing components. We hope that our approach accelerates the development of interpretable and efficient methods for diffusion-based text-to-image models.

ACKNOWLEDGEMENT

We would like to thank the Robert N. Noyce Trust for their generous gift to the University of California via the Noyce Initiative. The work was also partially funded by an unrestricted gift from Google and by the National Science Foundation award #2048122. The writers' opinions and conclusions in this publication are their own and should not be construed as representing the sponsors' official policy, expressed or inferred.

REPRODUCIBILITY STATEMENT

We release our core codebase containing the methodology implementation, settings, benchmarks containing compositional prompts under supplementary materials.

ETHICAL STATEMENT

As for the data collection and verification, we use the Amazon Mechanical Turk platform and form the comparison task as batches of HITs. We select workers from English-speaking countries, including the US, CA, UK, AU, and NZ, since the task require understanding the English input prompt. Each HIT takes around 15-30 seconds on average to accomplish, and we pay each submitted HIT with 0.15 US dollars, resulting in an hourly payment of 18 US dollars.

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

## A    RELATED WORK

**Text-to-Image Synthesis**    There are mainly three types of models for text-to-image synthesis: GAN-based (Tao et al., 2022; Zhu et al., 2019; Li et al., 2019; Fu et al., 2020; El-Nouby et al., 2019), autoregressive (Gu et al., 2022b; Lee et al., 2022; Ding et al., 2022) and diffusion models (Liu et al., 2021b; Nichol et al., 2021; Ruiz et al., 2022). Zhang et al. (2021) proposes XMC-GAN, a one-stage GAN that employs multiple contrastive losses between image-image, image-text, and region-token pairs. More recently, LAFITE (Zhou et al., 2022) enables language-free training by constructing pseudo image-text feature pairs using CLIP (Radford et al., 2021). As for autoregressive models, DALL-E adopts VQ-VAE to quantize image patches into tokens and then uses a transformer to generate discrete tokens sequentially (Ramesh et al., 2021). Parti (Yu et al., 2022) and Make-A-Scene (Gafni et al., 2022) both leverage classifier-free guidance to improve controllability. As for diffusion models, Gu et al. (2022a) concatenates VQ-VAE with the diffusion model and shows that the diffusion process can operate in discrete latent space. DALL-E 2 adopts the CLIP text encoder so that the diffusion process inverts the textual features into images (Ramesh et al., 2022).

**Structured Representations for Vision and Language**    Inferring shared structures across language and vision has been a long-term pursuit in unifying these modalities (Schuster et al., 2015; Johnson et al., 2018; Zhong et al., 2020; Lou et al., 2022). Wu et al. (2019) utilizes the structure from semantic parsing in a visual-semantic embedding framework to facilitate embedding learning. Wan et al. (2021) proposes a new task in which the goal is to learn a joint structure between semantic parsing and image regions. To the best of our knowledge, our work is the first attempt in T2I to incorporate language structures into the image synthesizing process.

**Diffusion Guidance**    To convert an unconditional diffusion model into a class-conditional one, Dhariwal & Nichol (2021) input the noisy image from each step into a classifier and calculate the classification loss. The loss can be back-propagated to the image space to provide a gradient that marginalizes the score estimation from the log of conditional probability. Similarly, in the T2I subdomain, Liu et al. (2021b) and Nichol et al. (2021) apply a noisy CLIP model to measure the cosine similarity between text prompts and noisy images.

## B    IMPLEMENTATION DETAILS

Throughout the experiments, we implement our method upon Stable Diffusion v1.4. For all comparisons between our method and Stable Diffusion, we fix the seed to generate the same initial Gaussian map and use 50 diffusion steps with PLMS sampling (Liu et al., 2021a). We fix the guidance scale to 7.5 and equally weight the key-value matrices in cross-attention layers if not otherwise specified. We do not add hand-crafted prompts such as "a photo of" to the text input. We use the Stanza Library (Qi et al., 2020) for constituency parsing and obtain noun phrases if not otherwise specified.

## C    VISUALIZATION OF ATTENTION MAPS

In this section, we demonstrate the visualization of cross-attention maps to support our assumptions and claims in Sec. 2. As is shown in Fig. 8, the attention maps of Stable Diffusion and our method

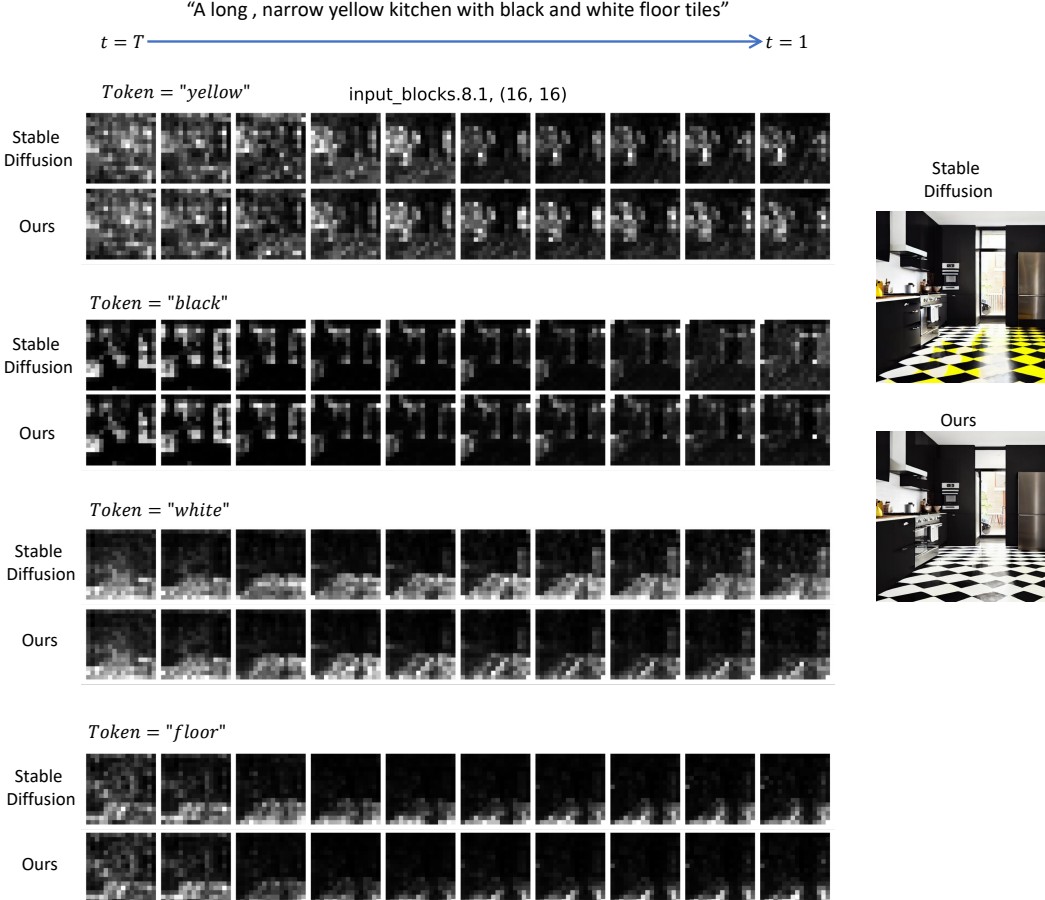

Figure 8: Visualization of cross attention maps of Stable Diffusion and our method. We compare maps of multiple tokens throughout the whole diffusion process with equal intervals.

have similar spatial distribution and highlights throughout the diffusion process. This phenomenon supports our assumption in Sec. 2.2 that the attention map $M_t$ is unchanged even with multiple values in each cross-attention layer. We can observe a similar phenomenon in Fig. 9 except that our method accelerates the formation of interpretable attentions for both "green" and "clock" tokens.

Fig. 8, 9 also justify our claim that values represent rich textual semantics mapped to the image space as contents. For instance, our method parses the prompt in Fig. 8 into "A long narrow yellow kitchen" and "black and white floor tiles", encodes and aligns them separately to form $\mathbb{V}$. Empirically, these operations enhance the semantics of "yellow" and "black and white" separately and mitigate "yellow" being blended into "black and white". This explains the disappearance of color leakage in our image compared to Stable Diffusion. Though one may attribute the leakage to incorrect attention distribution of the "yellow" token, we argue that this is not the critical reason. Despite the attention maps of "yellow" from our method slightly highlighting the "floor tile" regions, we cannot observe any yellow in our generated image. This proves that inaccurate attention distributions contribute little to the final image content. In addition, we also show in Fig. 10 that using multiple Keys is able to rectify the image layouts to mitigate missing object issues. The sheep-like attention maps in the third row verify the proposed variants of our method for concept conjunctions.

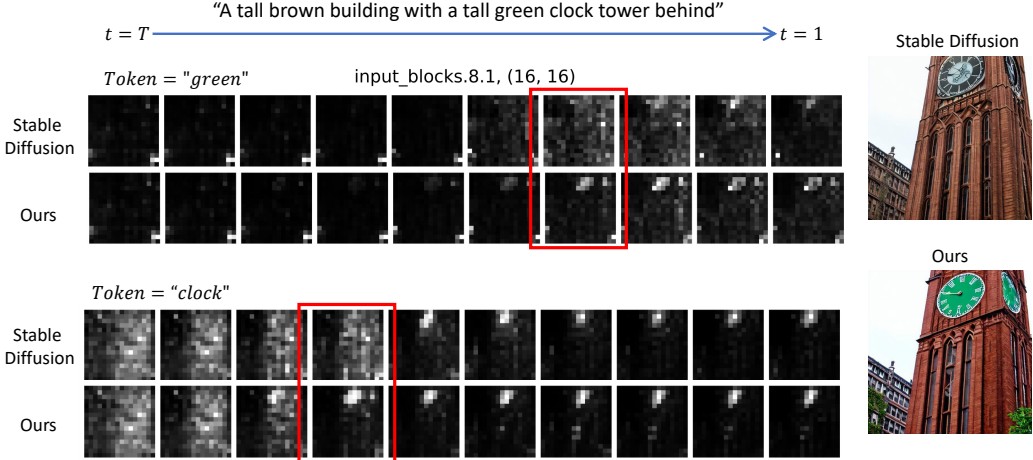

Figure 9: Visualization of cross attention maps corresponding to token "green" and "clock" across the full diffusion timestamps from step 50 to step 1 in equal intervals. Red boxes highlight steps where our method accelerates the formation of correct attention on the clock region. The evolution of the token "green" is also more interpretable in our method. Although the image composition is imperfect, the visualization still supports our assumptions and claims in Sec. 2.2.

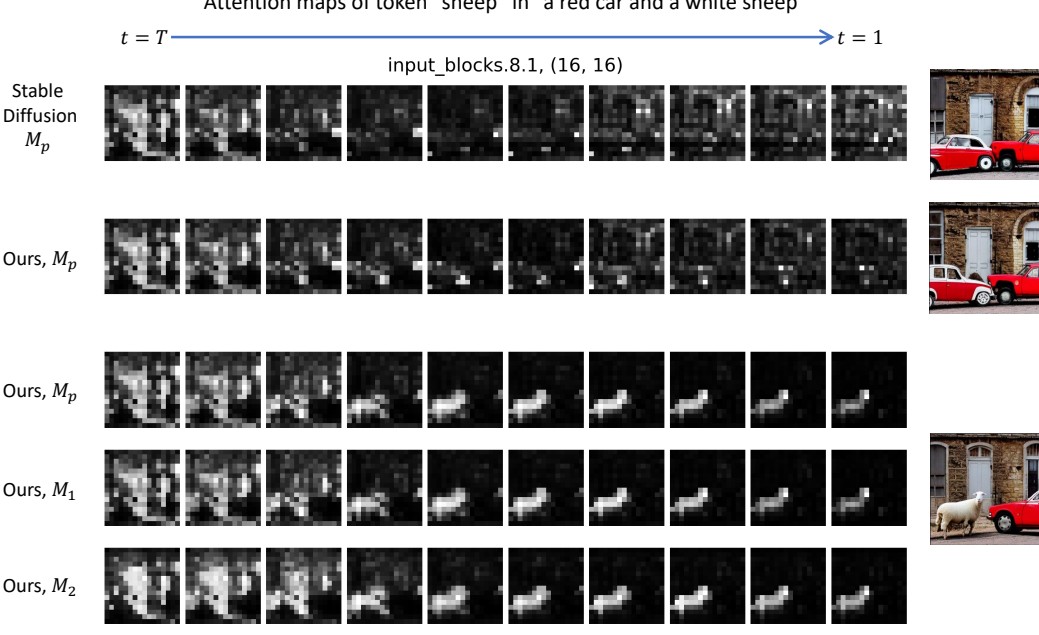

Figure 10: Visualization of attention maps for token "sheep" of different methods. Our method with multiple Keys successfully rectify image layouts.

# D ABLATION STUDY

## D.1 A CASE STUDY OF ATTRIBUTE BINDING

Here, we present a case study to show evidence of two root causes of incorrect attribute binding. The first one is the contextualized token embeddings due to causal attention masks. As is shown on the left side of Fig. 11, we first encode two different prompts with a shared component, e.g. "a red apple" as the naive one and "a green bag and a red apple". Using the encoding sequence of the naive prompt,

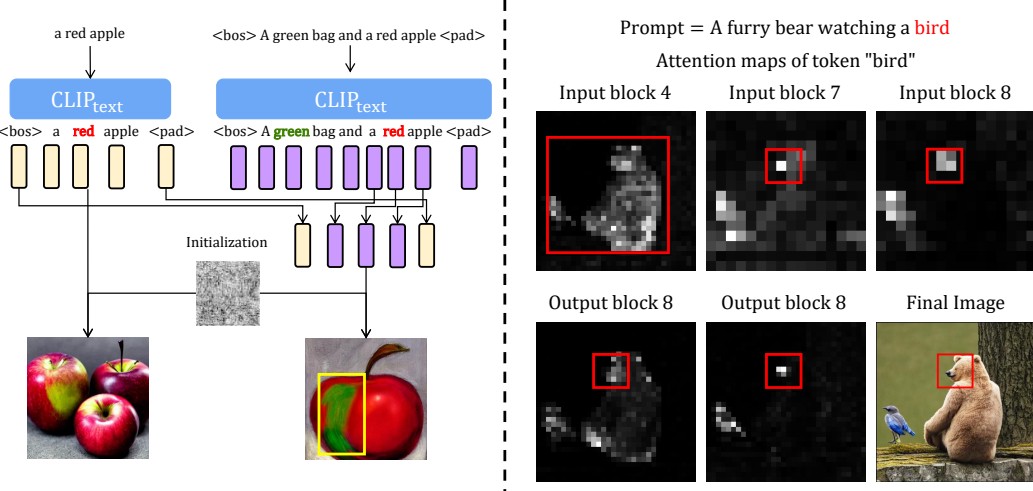

Figure 11: Examples showing the potential root causes of incorrect attribute binding. **Left**: The large green regions in the second image prove that the hidden state's output of token "red" is contextualized with token "green" before it. **Right**: Visualization of attention maps showing that the semantics from the token "bird" is mistakenly attended to the mouth region of the bear. The final image shows the unnatural beak-like shape of the bear.

we are able to get an image of red apple only. It is reasonable to assume that the yellow green regions are natural results of learning from authentic apple images. Then, we replace the tokens of the naive prompt with embeddings of the same token from the more complicated prompt. We use the same gaussian noise as initialization and generate an unnatural image with a solid green region (in the yellow bounding box). This result proves that the token "red" is contaminated with the semantics of "green" before it and explains some images with color leakage problems (e.g., Fig. 1).

The second reason attributes to inaccurate attention maps. In Fig. 11 (right), we visualize five cross-attention maps (averaged across attention heads) from both downsample and upsampling blocks. The attention maps show the salient regions corresponding to the token "bird". These maps demonstrate highlighted regions in the bottom left corner where the bird is located in the final image. Despite the interpretable structures, the maps also show saliency around the mouth region of the bear across all five layers. Thus, the inaccurate attention maps lead to a beak-like mouth of the bear in the image.

### D.2 COMPARISON OF PARSERS

In this subsection, we compare the difference between using a constituency parser and a scene graph parser to obtain text spans and generate images. Table 3 compares the extracted text spans using constituency parser and scene graph parser. Example 0 shows that both parsers end up with the same results for CC-500 prompts. For Example 1-4, the scene graph parser generates more spans than the constituency parser. We notice that concepts in the middle of the sentence appear more often in these spans than other noun tokens, like "egg" or "red sauce" in Example 3. This imbalance potentially explains why the "egg" looks more highlighted in Fig. 12 (bottom left). On the other hand, "orange slices" appear more often in constituency parsing results, leading to better "orange" textures in the generated image. Similar observations can be made in Example 2, where "green pole" is emphasized more often by the constituency parser.

## E LIMITATIONS & FUTURE WORK

There are several limitations of our work. First of all, our method depends on an external parsing function that may not be perfect. We adopt the commonly used Stanza Library Qi et al. (2020) for constituency parsing. The parsing function can be replaced with a more advanced learning-based method for improvement. Secondly, our method mainly focuses on compositional T2I neglecting any

| | Constituency Parser | Scene Graph Parser |
|---|---|---|
| Example 0 | CC-500 Prompt: *A white sheep and a red car* | |
| | "A white sheep", "a red car" | "A white sheep", "a red car" |
| Example 1 | Prompt: *A silver car with a black cat sleeping on top of it* | |
| | "A silver car", "a black cat", "A silver car with a black cat" | "A silver car", "a black cat", "top of it", "a black cat sleeping on top of it" |
| Example 2 | Prompt: *A horse running in a white field next to a black and green pole* | |
| | "A horse", "a white field", "a black and green pole", "a white field next to a black and green pole" | "A horse", "a white field", "a black and green pole", "A horse running in a white field" |
| Example 3 | Prompt: *Rice with red sauce with eggs over the top and orange slices on the side* | |
| | "red sauce", "the side", "the top and orange slices", "the top and orange slices on the side" | "red sauce", "the side", "the top and orange slices", "Rice with red sauce", "red sauce with eggs", "the top and orange slices on the side", "red sauce with eggs over the top and orange slices" |
| Example 4 | Prompt: *A pink scooter with a black seat next to a blue car* | |
| | "A pink scooter", "a black seat", "a blue car" | "A pink scooter", "a black seat", "a blue car", "a pink scooter with a black seat", "a black seat next to a blue car" |

Table 3: Comparison between the constituency parser and scene graph parser. For CC-500 prompts, both parsers end up with the same results. As for general prompts, scene graph parser tends to generate more text spans with middle concepts appearing multiple times across different spans.

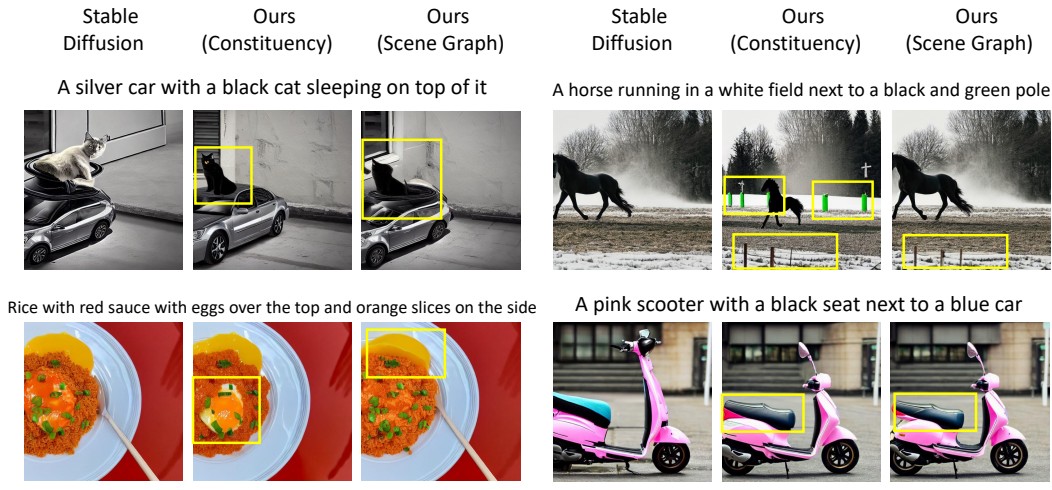

Figure 12: Synthesized images corresponding to prompts in Table 3. Yellow boxes annotate compositions that are improved using different parsers.

style descriptions. The parsing mechanism may categorize a style description, e.g. "in Van Gogh style" as a separate noun phrase that cannot be grounded in the image space. In addition, we discover that StructureDiffusion tends to generate similar images as Stable Diffusion. Thus we filtered out 20% of most similar image pairs in Table 1, considering the efficiency of human evaluation. Therefore, the improvement could be compromised when evaluated on the full set of generated images. Future work may focus on devising explicit methods to associate attributes to objects using spatial information as input. For example, how to make a text-to-image synthesis model interpret coordinate information with limited fine-tuning or prompt tuning steps would be an appealing direction.

## F  ADDITIONAL RESULTS

**Stable Diffusion**    **Ours**      **Stable Diffusion**    **Ours**

A green apple and a red banana

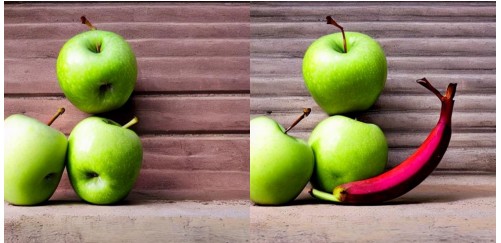

A red bird and a green banana

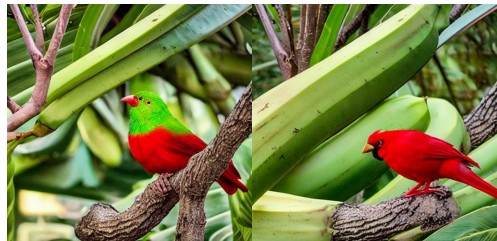

A gold clock and a green bench

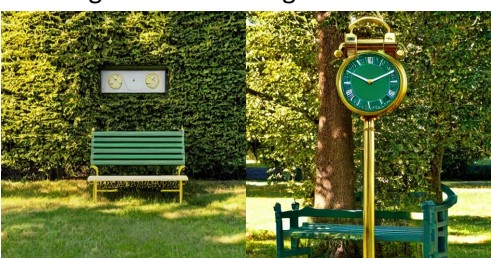

A blue bird and a brown bowl

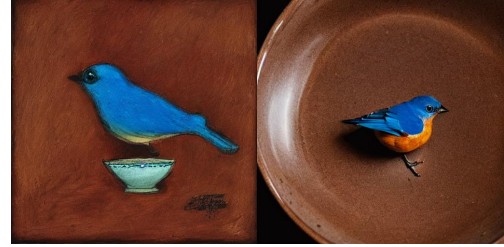

A red cake and a blue suitcase

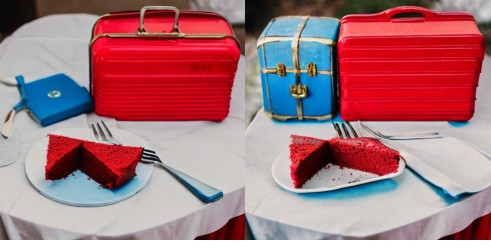

A blue backpack and a brown bear

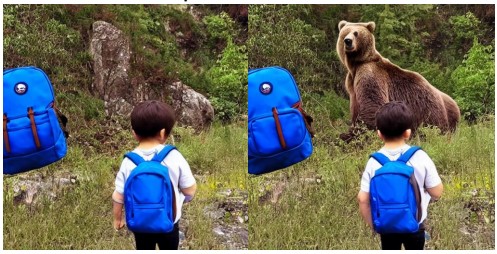

A brown dog and a blue suitcase

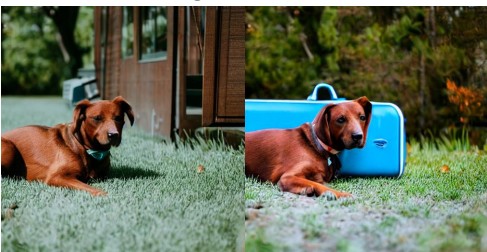

A blue backpack and a brown cow

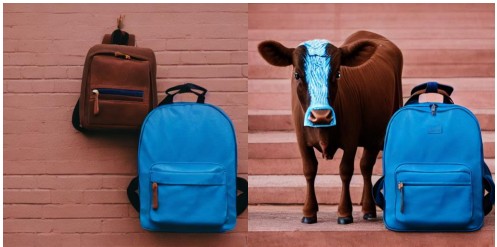

A yellow bowl and a blue cat

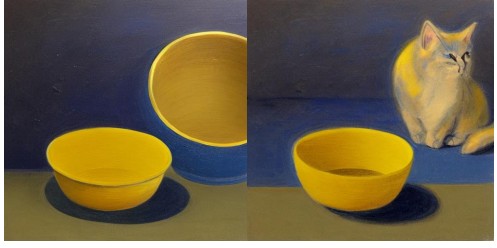

A brown dog and a blue horse

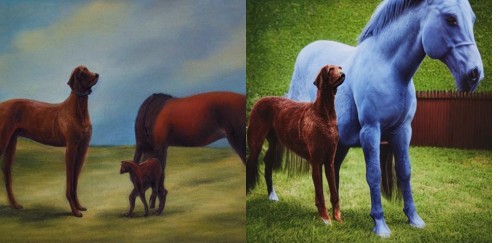

Figure 13: Qualitative results on CC-500

| Stable Diffusion | Ours | Stable Diffusion | Ours | Stable Diffusion | Ours |
|---|---|---|---|---|---|

a purple cat with a orange hat on its head

A red cat sits on a rug with a black cord

A yellow cat is wearing a blue plastic baseball hat.

A red helmet is on a yellow toilet in the dirt

A red stop sign above a white walk across road sign

Two elephants walking by a green wall with tan palm trees painted on it

A bathroom with red tile and a green shower curtain

A spacious kitchen has white walls , red countertops , and a large stove

A large white bed sitting in a hotel room next to a red couch

A white toilet bowl with a purple rug in front

A pink towel stands out greatly in the white bathroom

A large pizza on a white plate sitting on a blue table

A spoon and bowl of red pea soup and green beans with onions

A cow standing outside of a white building with a blue entrance

A black and white curtain hanging in a room that is decorated in black, white and red

Figure 14: Qualitative results on ABC-6K

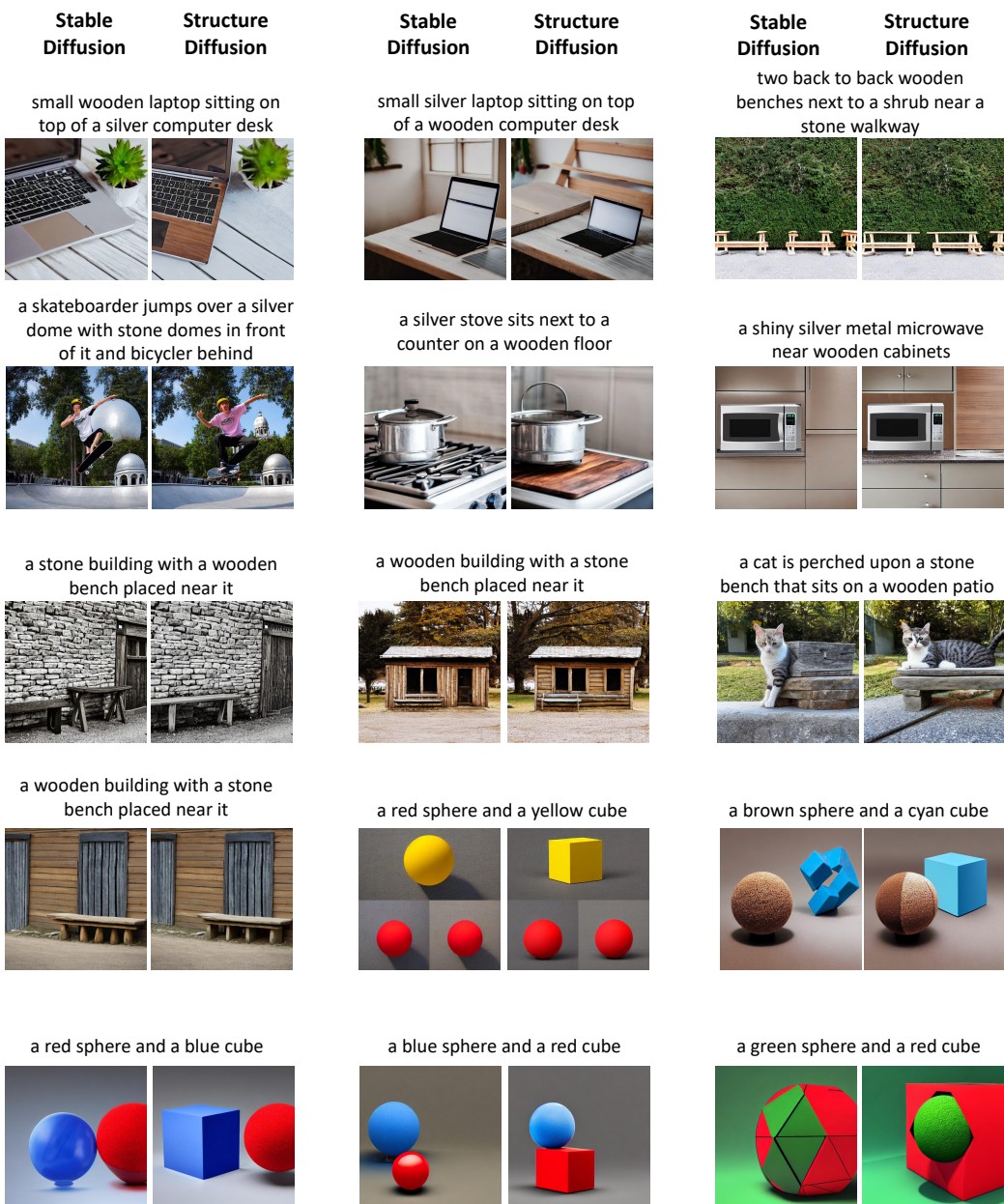

Figure 15: Qualitative results characterizing attributes beyond colors, including shape, size and materials.

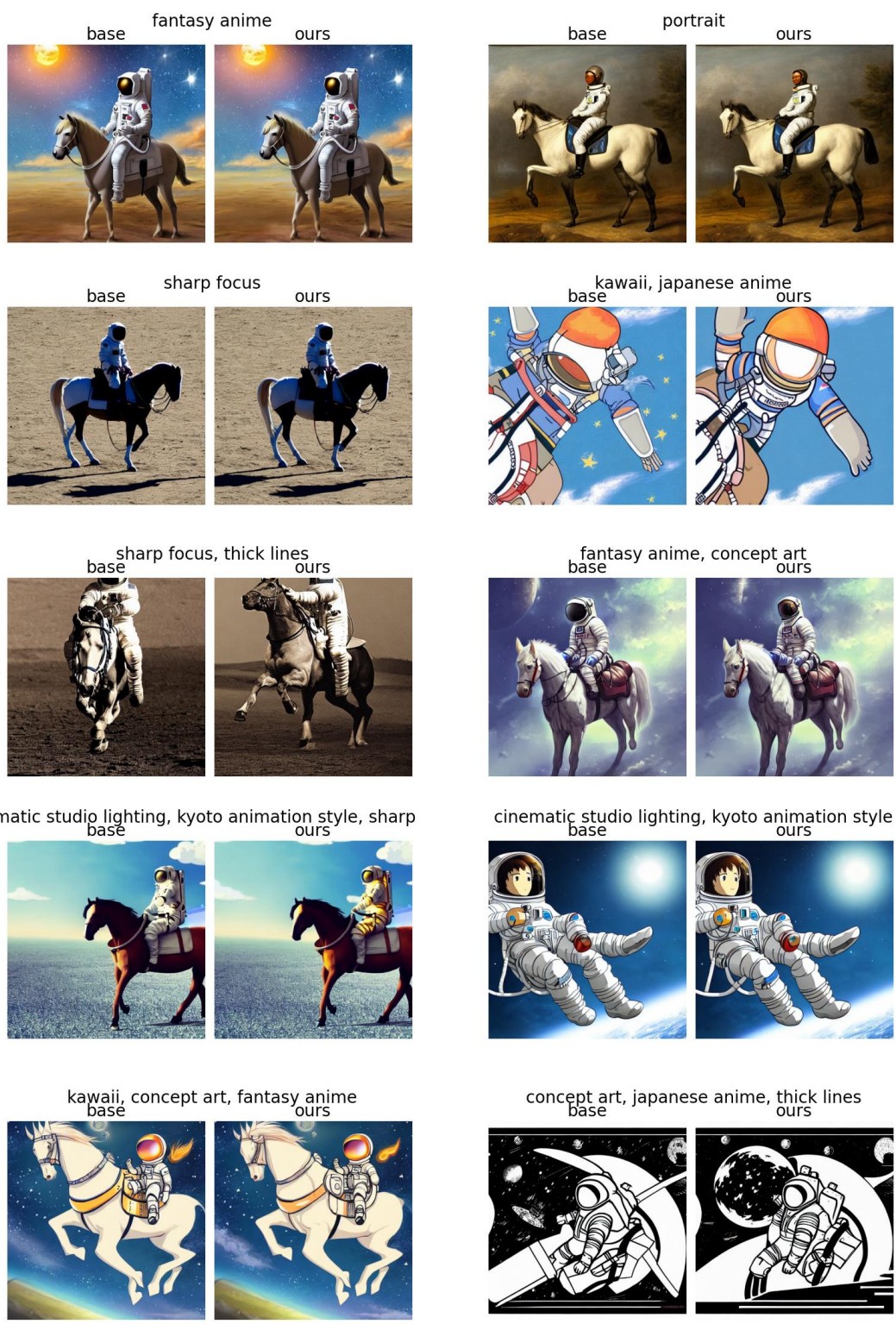

Figure 16: A prompt "an astronaut riding a horse" appended with different (combinations of) style descriptions. Our method has no negative effects on the image style. "base" refers to Stable Diffusion.

