# OpenReview forum: "Training-Free Structured Diffusion Guidance for Compositional Text-to-Image Synthesis"
_ICLR.cc/2023/Conference — ICLR 2023 poster_

### Official Review · Reviewer_mgR1 · 2022-10-22

**Confidence:** 4
**Correctness:** 3
**Technical Novelty And Significance:** 2
**Empirical Novelty And Significance:** 3
**Recommendation:** 6

**Clarity, Quality, Novelty And Reproducibility:**

The paper is clearly written. The proposed idea is original, but it is not clear to me how effective it is in solving the problems of attribute binding and especially compositionality.

**Strength And Weaknesses:**

Strengths:
1. The proposed idea of using the free supervision from structure in the prompts is interesting and worth investigating.
2. The problems of attribute-binding and composition in image generation are important and retain scope for improvement.
3. The authors propose two well-crafted datasets for evaluating the attribute-binding and compositional ability of image generation models and perform a comprehensive comparison of the proposed approach with baseline using a human study.
4. The paper is well written and organized, which makes it easy to follow and reproducible.

Weaknesses/suggestions:
1. It was not clear to me how human evaluators assess “image fidelity” in table 1. Please clarify this in the rebuttal.
2. It was not clear to me why the authors claim that  “[On CC-500] our method outperforms Stable Diffusion by around 5-8% in terms of both image-text alignment and image fidelity”, when the reported numbers in Table 1 show that StructureDiffusion wins against Stable Diffusion (SD) only 31.8% of the time, while SD wins 38.9%. These numbers make me conclude that SD is actually superior to StructureDiffusion for compositionality. Please clarify this in the rebuttal.
3. If I am interpreting the results in Table 1 correctly, then  StructureDiffusion is 20% better than SD in alignment, but 8% worse in compositionality. These results are not discussed in detail in the paper, which could help us understand how the proposed approach is helping in these tasks.
4. No qualitative experiments are performed to show how the weighed cross-attention maps affect image formation (see the Hertz et al. paper for examples, which is used as an inspiration for the proposed approach) during the diffusion process.


**Summary Of The Paper:**

This paper proposes a method for improving the attribute binding and compositional ability of denoising diffusion probabilistic models for image generation. The authors utilize the fact  that the attribute-object relation pairs can be obtained as text spans for free from the parsing tree of the sentence used for conditioning image generation. Using this free source of information, the authors propose  a method to combine the structured representations of prompts with the diffusion guidance process. Specifically, using the text spans from the parsing tree, the method obtains a cross-attention output that is a weighted average of the attention map of the full prompt multiplied by the value tensor of the full prompt and the value tensors of individual noun-pairs obtained by the parsing tree. To evaluate the method, the authors create two datasets. The first dataset, Attribute Binding Contrast set (ABC-6K) evaluates attribute binding ability of the method, by creating prompts with common and uncommon colors binding to objects in the COCO dataset. The second dataset, Concept Conjunction 500 (CC-500), tests the model’s ability to compose together objects that rarely appear together in reality. Pairwise comparisons in human evaluation study are used to compare the method to the baseline model (Stable diffusion) and to Composable diffusion for the CC-500 dataset. The proposed approach, StructureDiffusion, is preferred by users over Stable Diffusion on ABC-6K, while it performs worse on CC-500. However, StructureDiffusion outperforms Composable Diffusion on CC-500. FID scores on COCO are comparable to original Stable Diffusion.

**Summary Of The Review:**

I am on the fence about how effective the proposed approach is in solving the problems of attribute binding and especially compositionality. It is also not clear to me why and how the proposed approach is working in different cases. So currently, I consider the paper to be borderline, but I am happy to re-evaluate after the rebuttal stage if I get clarification on the issues raised above.

---

> ### Author Response · Authors · 2022-11-15
> **Response to Official Review by Reviewer mgR1**
>
> Thank you for your valuable feedback.
>
> **We want to clarify a critical typo made in Table 1. We accidentally swapped the headers “Lose” and “Tie” columns when we converted the presentation from bar plots to a table. This caused a mismatch between the table and the descriptions in sections 3.2 - 3.3. It may have caused some misunderstanding regarding the models’ performance.**
>
> >Weaknesses 1: It was not clear to me how human evaluators assess “image fidelity” in table 1.
>
> We ask annotators, “Regardless of the text, which image is more realistic and natural”. In the instructions, we ask annotators to ignore the text description and focus on details that make the image look more realistic than the other. We have supplemented this detail in Sec. 3.1 in the revision.
>
>
> >Weaknesses 2: These numbers make me conclude that SD is actually superior to StructureDiffusion for compositionality.
>
> Please see the beginning of our responses regarding the typo in Table 1. For CC-500, we add an experiment asking annotators to answer whether an object mentioned in the prompt appears in the generated image. We also attempt an automatic evaluation using GLIP, a SOTA phrase grounding model. Overall, our method improves by 3.5% in generating complete objects with correct colors. Please refer to Table 2 and Sec. 3.3 for more details.
>
>
>
> >Weaknesses 3: If I am interpreting the results in Table 1 correctly, then StructureDiffusion is 20% better than SD in alignment, but 8% worse in compositionality.
>
> Both ABC-6K and CC-500 evaluate the image-text alignment with a focus on compositionality. StructureDiffusion is 4.1%-6.6% better than SD in ABC-6K and CC-500. We further quantify the “compositionality” for CC-500 in Table 2 and Sec. 3.3
>
>
> >Weaknesses 4: No qualitative experiments are performed to show how the weighted cross-attention maps affect image formation
>
> We have added a section in Appendix C with multiple figures to illustrate the evolution of attention maps throughout the diffusion process. Please refer to Appendix C and Fig. 8-10. Qualitative examples (Fig. 8,9) show that the attention maps from the full sentence have no change or only slightly change when we apply multiple values in the diffusion process. When multiple keys are applied to obtain, the attention maps of NPs ($M_i$) look very similar to maps of the full sentence ($M_p$), except that $M_i$ have stronger and larger highlighted regions around the object. For example, in Fig. 10, $M_2$, corresponding to “a white sheep” enhanced, shows larger and brighter attention on the sheep. $M_2$ has an average value of 3.3e-2 compared to 2.6e-2 for $M_p$ and $M_1$, which means maps created from NP “a white sheep” have higher response to the visual “sheep” region.

---

> > ### Author Response · Authors · 2022-11-19
> > **Follow-up at the end of stage 1**
> >
> > Thank you again for your efforts and valuable feedback. **We kindly invite you to check out our revised paper with more experimental results and comprehensive visualizations and examples.** Please let us know if we have addressed all your concerns in our response.
> >
> > In addition, we would like to elaborate more on the following point:
> >
> > >Weaknesses 3: If I am interpreting the results in Table 1 correctly, then StructureDiffusion is 20% better than SD in alignment, but 8% worse in compositionality.
> >
> > To have more quantified results, we conduct an additional evaluation on CC-500 (Table 2, Sec. 3.3). We ask human annotators to annotate whether every concept in the prompts appears in the image. We also evaluate our images using GLIP, the SOTA model that grounds phrases into bounding boxes. Human indicates a 3.5% improvement in generating complete compositions with correct colors, while GLIP indicates a 0.2% improvement. We observe that GLIP fails to ground colors correctly and align with human annotations only 48% of the time(https://imgur.com/a/UX0D1s1). Compositional understanding remains a challenge for vision-language models (Thrush et al., 2022); thus, compositionality evaluation is also an open research field. We believe a better vision-language model in the future can bring a more precise evaluation of compositionality.

---

> > > ### Comment · Reviewer_mgR1 · 2022-12-11
> > > **Rebuttal response**
> > >
> > > Thanks you for your detailed response to my reviews and the additional experiments --- they have addressed my questions and I have raised the score for the paper.

---

### Official Review · Reviewer_QzN2 · 2022-10-24

**Confidence:** 4
**Clarity, Quality, Novelty And Reproducibility:** The quality of this paper is good, wi…
**Correctness:** 3
**Technical Novelty And Significance:** 3
**Empirical Novelty And Significance:** 3
**Recommendation:** 6

**Details Of Ethics Concerns:**

The manipulation technique can cause the problem of privacy.

**Strength And Weaknesses:**

Strength:
This paper proposes to manipulate the cross-attention representations based on linguistic insights. And the compositional performance can be obviously improved.

Weakness:

1. As shown in Figs. 4 and 5, the proposed diffusion model's effects are mainly observed with the color attributes. The authors should provide the comparison with more types of attributes to demonstrate the improvement of compositional skills.

2. Why the quantitative comparisons between Stable Diffusion and Composable Diffusion are not conducted with the metrics of IS, FID, and R-prec?

3. The authors have claimed that their method can also take the scene graphs as the inputs. They should also compare the results with existing scene-graph to image approaches. And the visual cases in Fig. 6 have some errors. For example, there is not beach in the middle column of Fig. 6, while beach exists in the scene graph.


**Summary Of The Paper:**

This paper proposes a new diffusion-based T2I models to improve the compositional skills. The linguistic structures are incorporated with the diffusion guidance process based on the controllable properties of manipulating cross-attention layers in diffusion-based T2I models. The model is built based on the stable diffusion, and the proposed cross-attention design requires no additional training samples. The model can achieve SOTA compositional skills in qualitative and quantitative results.

**Summary Of The Review:**

I think the proposed linguistic insights strategy is a useful trick for T2I diffusion models. However, the experimental results are not convincing enough.

---

> ### Author Response · Authors · 2022-11-15
> **Response to Official Review by Reviewer QzN2**
>
> Thank you for your valuable feedback.
>
>
> >Weaknesses 1: The authors should provide the comparison with more types of attributes to demonstrate the improvement of compositional skills.
>
> Thanks for the suggestions. We manually collect a set of 100 prompts from multiple datasets like MSCOCO, CLEVR, and Drawbench, characterizing attributes like shape, size, and materials. We show the qualitative results in Fig. 15. Our method can improve attribute binding beyond colors. We also obverse that color is the most erroneous attribute, and missing object is potentially the most critical issue of general compositionality.
>
> >Weaknesses 2: Why the quantitative comparisons between Stable Diffusion and Composable Diffusion are not conducted with the metrics of IS, FID, and R-prec?
>
> As is discussed in Sec. 3.1, IS and FID measures image quality and diversity. These metrics cannot reflect the performance in generating correct compositions. R-Prec measures image-text alignment. However, it measures scene-level similarity and cannot reflect compositional performance. CLIP is particularly fragile in compositional understanding, as indicated in previous work (Thrush et al., 2022). Hence, We did not report these metrics to avoid misunderstanding our goal of improving compositionality instead of quality. Nevertheless, we show these metrics for reference.
>
>
> |                | IS   | FID  | R-prec |
> |----------------------|------|------|--------|
> |ABC-6K||||
> | Stable Diffusion     | 21.5 | 21.7 | 74.0   |
> | Ours                 | 21.4 | 21.7 | 73.0   |
> | CC-500               |      |      |        |
> | Stable Diffusion     | 16.5 | -    | 68.5   |
> | Composable Diffusion | 14.3 | -    | 69.7   |
> | Ours                 | 16.9 | -    | 67.8   |
>
>
> >Weaknesses 3: Scene graphs
>
> Please note that our scene graph setting is not a scene graph-to-image model but a text-to-image model. We only utilize the scene graph parser to obtain a structured representation of a text prompt and extract spans from it. Therefore, we do not compare it to other scene graph-to-image models. The “beach” node is a typo due to copying the graph in the left example. We have corrected it in our revision. We also comprehensively evaluate and analyze the scene graph setting in Sec. 3.4 and Appendix D.2
>
> >The experimental results are not convincing enough.
>
> To quantify our evaluation beyond human judgment, we conducted an additional experiment on CC-500, asking annotators to answer whether an object mentioned in the prompt appears in the generated image. We also attempt an automatic evaluation using GLIP, a SOTA phrase grounding model. Please refer to Table 2 and Sec. 3.3 for more details. We have also added visualization and analysis on the attention maps in Appendix C and D.2 to verify our assumptions and claims made. We have also supplemented more visualization and analysis on the evolution of cross-attention maps throughout the diffusion process in Appendix C. Additional qualitative results are provided in Fig. 12, 14-16.

---

> > ### Author Response · Authors · 2022-11-19
> > **Follow up at the end of stage 1**
> >
> > Thank you again for your efforts and valuable feedback. **We kindly invite you to check out our revised paper with more experimental results and comprehensive visualizations and examples.** Please let us know if we have addressed all your concerns in our response.
> >
> > In addition, we would like to elaborate more on the following point:
> >
> > > Weaknesses 1: The authors should provide the comparison with more types of attributes to demonstrate the improvement of compositional skills.
> >
> > Our method improves Stable Diffusion from multiple aspects. In addition to attributes, we asked users to indicate why our images are better than the Stable Diffusion ones in ABC-6K. 31% voted for “fewer missing objects”, 14.1% voted for “better colors”, and 54.8% voted for “other attributes or details”. Therefore, our improvement takes place in multiple aspects, including attributes, missing objects, and fine-grained details.

---

### Official Review · Reviewer_ZtBV · 2022-10-25

**Confidence:** 4
**Correctness:** 3
**Technical Novelty And Significance:** 3
**Empirical Novelty And Significance:** 3
**Recommendation:** 6

**Clarity, Quality, Novelty And Reproducibility:**

The paper is well written, although I believe the notation of the cross-attention descriptions can be improved (e.g. $l$ and $d$ in Fig 2 do not seem to align with equations) and described more intuitively.
The approach itself seems novel to me.

**Strength And Weaknesses:**

Previous work has shown that the cross-attention layers have a lot of influence on the final content of generated images and, therefore, it makes sense to look into those more closely in this respect. The proposed approach does not need any additional training data or training of the model, making it easy to use in practice. The qualitative evaluation shows some improvement over the baseline model (Stable Diffusion) and human user studies indicate that the generated images are indeed better.

Some questions and concerns:
- I am not sure about the human user study; the authors say that they generate images from both approaches (their approach and the baseline Stable Diffusion model), then remove 20% of the most similar images, and only then run the user study; I understand that a lot of the images are similar since the same seed is used, but given that the performance of both approaches is probably similar on those 20% of the images this will affect the final results; in reality, the performance of the two models may be much closer to each other than the human user study indicates
- Table 1, CC-500, comparison to SD seems that SD is actually better, but the text seems to indicate otherwise?!
- Does it scale to longer captions with many constituents?
- Also, looking at real-life use people seem to append a lot of words for specific styles, how would behavior like that affect the performance? Captions like *a photograph of an astronaut riding a horse portrait pixiv art soft lines fantasy anime smooth illustration murata range 8 k studio ghibli sharp focus wlop kyoto animation style art viewer anime art close sharp detailed pixiv kawaii concept art high quality cute sunset exile intricate hd elegant 4 k cute face artgerm madhouse stunning ufotable valley alphonse mucha detailed background anime portrait yoshinari yoh artgem detailed face japanese anime cinematic studio lighting pink hue white background thick lines large eyes*

**Summary Of The Paper:**

The paper proposes a way to improve compositional text-to-image in large, pre-trained diffusion models without additional data or training requirements. To achieve this, language parsers are used to extract high-level information from a given caption (e.g. noun phrases) and use those to obtain additional text embeddings. These text embeddings are then used in combination with the original ones to scale the outputs of cross-attention layers. Experiments show that the approach can improve attribute binding and compositionality in generated images.

**Summary Of The Review:**

Overall I think the approach is novel, may increase the performance in some specifc settings, and does not need any additional training or data. I'm not entirely sure about the evaluation with the human user study and how well the approach will scale to more complex/longer captions or the kinds of captions that users seem to use in reality.
However, I see value in the approach and its simplicity and easy applicability are advantageous.

---

> ### Author Response · Authors · 2022-11-15
> **Response to Official Review by Reviewer ZtBV**
>
> Thank you for your valuable feedback.
>
>
> **We want to clarify a critical typo made in Table 1. We accidentally swapped the headers "Lose" and "Tie" when we converted the presentation from bar plots to a table. This caused a mismatch between the table and the descriptions in sections 3.2 - 3.3. It may have caused some misunderstanding regarding the models’ performance.**
>
>
> >Question 1: I am not sure about the human user study
>
> If we consider the 20% images resulting in “Tie” in comparison, the lower bound of our performance comes to 4-6.4%. Considering that our method is “training-free”, our method is indeed effective, and the improvement is reasonable. To further substantiate our improvement, we conducted a more objective experiment on CC-500, asking human annotators to answer whether an object in the prompt appears in the generated image (see Sec. 3.3 and Table 2). We also applied GLIP as an automatic metric as a comparison. As a result, our method improves 3.5% (relatively 18%) of the images to have complete objects with correct colors and outperforms Composable Diffusion. It is important to note that these prompts are pretty challenging for Stable Diffusion, making our improvement significant in relative numbers.
>
> However, our method is effective and novel in that it attempts to blend the structured representations from the language side into the multimodal interaction stage based on the intriguing properties of cross-attention guidance in diffusion T2I models.
>
>
> >Question 2: Table 1, CC-500, comparison to SD seems that SD is actually better, but the text seems to indicate otherwise?!
>
> Please see our bolded text above for the typo issue in Table 1. We are regretful that this typo has caused misunderstanding during the review process.
>
>
> >Question 3: Does it scale to longer captions with many constituents?
>
> Yes, our method works with many constituents. Note that ABC-6K already has prompts containing a range of 4-13 constituents. We supplement some qualitative results in Fig. 14 of prompts that consist of 3-6 noun phrases. We have also tested on 22 incredibly long prompts from Drawbench with an average of 30 tokens (~8 noun tokens/prompt). We did not notice significant differences between our method and Stable Diffusion in this toy experiment (https://imgur.com/a/mNl2z8H). Yet, we think it sufficient to show that our method can be applied to long prompts.
>
>
>
> >Question 4: Also, looking at real-life use people seem to append a lot of words for specific styles, how would behavior like that affect the performance?
>
> Thanks for the great question. The provided prompt is too long for the text encoder. We also observe marginal gain as multiple different style descriptions are appended together. Therefore, we verify our method with one/two/three styles concatenated after “an astronaut riding a horse” (Fig. 16) or “a red car and a white sheep” (https://imgur.com/a/B8OBV5o). We generally do not observe any corruption and our model still improves the compositionality. For application purposes, we may ask users to select words that are style so that our method will not extract them as NPs.

---

> > ### Author Response · Authors · 2022-11-19
> > **Follow up at the end of stage 1**
> >
> > Thank you again for your efforts and valuable feedback. **We kindly invite you to check out our revised paper with more experimental results and comprehensive visualizations and examples.** Please let us know if we have addressed all your concerns in the response.

---

### Official Review · Reviewer_5ZP1 · 2022-10-26

**Confidence:** 5
**Correctness:** 4
**Technical Novelty And Significance:** 2
**Empirical Novelty And Significance:** 3
**Recommendation:** 6

**Clarity, Quality, Novelty And Reproducibility:**

(+) The human evaluation protocol is clearly described.

(-) Abstract and intro
- we observe that attribution-binding and compositional capabilities are still considered major challenging issues. <- The readers cannot guess what attribution-binding and compositional capability from abstract.
- Abstract has peripheral aspects (e.g., cross-attention) instead of the key component (tokenizing sentences).

(-) What is "functionality between attention maps and token semantics"?

(-) t and t_i are confusing.

(-) Eq. 4 maintains the image layout  <- What is the original image layout? Maybe the image layout from a full sentence embedding?

(-) Assuming above guess is correct, why does Mt remain unchanged although the sentence embedding and the token embeddings are different?

(-) c denotes hierarchies but NPs in a same hierarchy have different cs in Figure 3.

typo
- downsample and upsampling blocks -> downsampling and upsampling blocks


**Strength And Weaknesses:**

Strengths

(+) The proposed method does not require training samples.

(+) This paper introduces a benchmark dataset to evaluate attribute binding.

(+) The experiments visualizes the causes of incorrect attribute binding (Figure 8).

(+) Sec. 2.1 provides thorough background.

Weaknesses

(-) Missing analysis: How do the attention maps from NPs and sentence evolve throughout the timetsteps?

(-) The contributions are limited.
- The attributes are limited to colors.
- The authors do not provide comparisons in the scene graph setting.
- Evaluation of the compositionality is limited to user study.
- Preventing objects from missing is not evaluated.


**Summary Of The Paper:**

This paper proposes a modification of stable diffusion to better reflect per-object attributes in a sentence to the corresponding objects in the image and prevent missing objects.
For example, feeding "a red car and a white sheep" into stable diffusion produces and image with a red car and a red sheep.
For experiments, the authors introduce a new dataset for evaluation.

**Summary Of The Review:**

I value the colors being correct in the resulting images. However, contribution of this paper is limited as mentioned in weaknesses. I consider the bar for ICLR is much higher.

---

> ### Author Response · Authors · 2022-11-15
> **Response to Official Review by Reviewer 5ZP1 (Cont'd)**
>
> >Clarity 1: Abstract and Intro.
>
> We have addressed the definition of attribute binding and compositionality in the revision.
> Cross-attention is not a peripheral aspect of our method, and we think it essential to bring up the idea in the abstract.
> Tokenizing sentences is not a key component of our method.
>
>
>
> >Clarity 2: What is "functionality between attention maps and token semantics"?
>
> We were referring to the claim that image layout depends on attention maps while content relies more on values computed from token embeddings. We have revised the paragraph accordingly.
>
>
> >Clarity 3: t and t_i are confusing.
>
> We have removed t_i and revised equation 1-6 accordingly.
>
>
> >Clarity 4: Eq. 4 maintains the image layout <- What is the original image layout? Maybe the image layout from a full sentence embedding?
>
> The “original layout” refers to the layout when only a single value (i.e., $f_V(\mathcal{w}_p)$) is applied to cross attention. So yes, it refers to the image layout from a full sentence embedding in Stable Diffusion. We have revised the wording to make it clear.
>
>
> >Clarity 5: Assuming above guess is correct, why does Mt remain unchanged although the sentence embedding and the token embeddings are different?
>
> Even though we have $\mathcal{W}_1, \mathcal{W}_2, \ldots$ for multiple values, $M_t$ is still calculated from $\mathcal{W}_p, Q^t$ only. Therefore, we hypothesize that having multiple values only affects image content but not layout, which implies that $M_t$ is unchanged. We have also verified this hypothesis in Appendix C and Fig. 8-10
>
>
> >Clarity 6: c denotes hierarchies but NPs in a same hierarchy have different cs in Figure 3.
>
> $\mathcal{C}$ denotes a set of NPs from different hierarchies while $c_i$ denotes an NP from a specific hierarchy.

---

> > ### Author Response · Authors · 2022-11-19
> > **Follow up at the end of stage 1**
> >
> > Thank you again for your efforts and valuable feedback. We kindly invite you to check out our revised paper with more experimental results and comprehensive visualizations and examples. Please let us know if we have addressed all your concerns in our response.
> >
> > In addition, we would like to elaborate more on the following point:
> >
> > >Weaknesses 1: The attributes are limited to colors.
> >
> > The improvement from our method goes beyond colors. As mentioned in the previous response, we run a small set of prompts to characterize attributes in shape, size, and materials. Our method is effective beyond color, as shown in Fig. 15. To further validate our argument, we asked users to indicate why our images are better than Stable Diffusion ones. 31% voted for “fewer missing objects”, 14.1% voted for “better colors”, and 54.8% voted for “other attributes or details”. Therefore, our method improves in multiple ways, including all kinds of attributes, missing objects and fine-grained details.

---

> > ### Comment · Reviewer_5ZP1 · 2022-11-19
> > **Thank you**
> >
> > Thank you for the clarification. Please check the remaining concerns.
> >
> > **Tokenization and cross-attention**
> >
> > I agree that the observation in the cross-attention layers is important but not as important as linguistic structures and realigning part. Linguistic structures are also considered important by other reviewers. Without it, NPs do not exist and thus structured diffusion guidance cannot exist. Furthermore, I think the realigning part is the main of structured diffusion guidance.
> >
> > **Contribution**
> >
> > I still think that the contribution is limited. The authors argue that the two main contributions in the paper are 1) discovering the weaknesses of Stable Diffusion regarding compositionality and 2) designing structured guidance.
> >
> > 1) The weakness regarding compositionality is (was) supported by only some examples. For rigorous argument, there should have been more scientific or objective evaluation protocols. I am not sure whether the revision is resolving concerns with minor changes or it is a significant change that adds more contribution. I think if we accept this due to the additional results and evaluations in the revision, future submissions may do the same. Maybe it would be fine if this is a journal paper under major revision. But it is like extending the submission deadline.
> >
> > 2) The weakness the authors fixed lies in the representation of the conditions, not in the stable diffusion. The structured guidance introduces bindings in the language representation to reflect the linguistic structures. I think the storyline of the paper is a bit misleading.

---

> ### Author Response · Authors · 2022-11-15
> **Response to Official Review by Reviewer 5ZP1**
>
> Thank you for your valuable feedback.
>
> >Weaknesses 1: Missing analysis
>
> We have added a section in the appendix with multiple figures to illustrate the evolution of attention maps throughout the diffusion process. Please refer to Appendix C and Fig. 8-10. Qualitative examples (Fig. 8,9) show that the attention maps from the full sentence have no change or only slightly change when we apply multiple values in the diffusion process. When multiple keys are applied, the attention maps of NPs ($M_i$) look very similar to maps of the full sentence ($M_p$), except that $M_i$ have stronger and larger highlighted regions around the object. For example, in Fig. 10, $M_2$, shows larger and brighter attention on the sheep. $M_2$ has an average value of 3.3e-2 compared to 2.6e-2 for $M_p$ and $M_1$, which means maps created from NP “a white sheep” have higher response to the visual “sheep”.
>
> >The contributions are limited
>
> We respectfully disagree. Our method improves the image compositionality from multiple aspects and is beneficial for the community to understand the importance of text encoders and individual tokens in the diffusion process. Please see our responses below and general response.
>
>
>
> >Weaknesses 2-1: the attributes are limited to colors.
>
> We manually collect a set of 100 prompts from multiple datasets like MSCOCO, CLEVR, and Drawbench, characterizing attributes like shape, size, and materials. We show the qualitative results in Fig. 15. Our method can improve attribute binding beyond colors. Note that our method also mitigates missing objects and other fine-grained details.
>
> >Weaknesses 2-2: the authors do not provide comparisons in the scene graph setting
>
> In the revised paper, we include detailed evaluation of scene graph setting (Sec. 3.5) and comprehensive comparison between scene graph and constituency parser (Appendix D.2). For ABC-6K, our scene graph setting achieves 34.2%-32.9% Win-Lose rate when compared to Stable Diffusion (similar results in alignment and fidelity). In general prompts from MSCOCO, it achieves 39.2 IS, 17.9 FID, and 72.0 R-Precision, maintaining the overall image quality. We observe that the scene graph parser tends to generate more text spans (4.8 spans/prompt) than the constituency parser (4.5 spans/prompt). Concepts in the middle of the sentence appear more frequently than other concepts. For example, noun tokens appear in an average of 2.04±1.0 spans for the scene graph parser while 1.78±0.87 spans for the constituency parser. We conjecture that this imbalance marginalizes the improvement in the scene graph setting.
>
> >Weaknesses 2-3: Evaluation of the compositionality is limited to user study
>
> To have more quantified results, we conduct an additional evaluation on CC-500 (Table 2, Sec. 3.3). We ask human annotators to annotate whether every concept in the prompts appears in the image. We also evaluate our images using GLIP, the SOTA model that grounds phrases into bounding boxes. We observe that GLIP results differ greatly from human annotations, making it less reliable in compositionality evaluation. We observe that GLIP fails to ground colors correctly (https://imgur.com/a/UX0D1s1). Compositional understanding remains a challenge for vision-language models (Thrush et al., 2022); thus, compositionality evaluation is also an open research field. We believe a better vision-language model in the future can bring a more precise evaluation of compositionality
>
> >Weaknesses 2-4: Preventing objects from missing is not evaluated.
>
> Please see our response to weaknesses 2-3 and Table 2 in the paper. Overall, our method improves 3.5% of the images from missing objects in CC-500 over Stable Diffusion and also outperforms Composable Diffusion by 2.1%.

---

> > ### Comment · Reviewer_5ZP1 · 2022-11-19
> > **Thank you**
> >
> > Thank you for the effort in the response and revision.
> >
> > More experiments support that the proposed method generalizes to shapes and materials but not to sizes: only one successful sample with `small` among two `small`s and three `large`s.
> >
> > Experiments with scene graph and missing objects are reinforced.
> >
> > Using GLIP for quantitative evaluation is meaningful.
> >
> > I am not sure this many additional results in the rebuttal process is okay to support the contributions. To me, it is adding contribution after the submission which is against the policy.

---

### Public Comment · ~Mehmet_Ozgur_Turkoglu1 · 2022-11-07
**provided code is not working!**

Thanks for the interesting work. I spent quite few time making the code run but the provided code does not work. It does not have a README file and can not be run readily. For instance, even the default parameter for structural attention is set to None as follows (txt2img.py, line 399).

    parser.add_argument(
        "--struct_attn",
        type=str,
        choices=['none', 'extend_str', 'extend_seq', 'align_seq'],
        default='none'
    )


Text conditioning (c) returns a list as follows (txt2img.py, line 562):

                            if opt.struct_attn == 'extend_str':
                                # repeat each NP in string to expand the embed seq
                                for i in range(1, len(nps)):
                                    nps[i] = (" " + nps[i]) * (model.cond_stage_model.max_length // len(nps[i].split()))
                                nps = [[np]*len(prompts) for np in nps]
                                c = [model.get_learned_conditioning(np) for np in nps]


This c goes to a sampler (e.g plm_sampler) as follows:

https://github.com/CompVis/stable-diffusion/blob/69ae4b35e0a0f6ee1af8bb9a5d0016ccb27e36dc/ldm/models/diffusion/plms.py#L82

        if conditioning is not None:
            if isinstance(conditioning, dict):
                cbs = conditioning[list(conditioning.keys())[0]].shape[0]
                if cbs != batch_size:
                    print(f"Warning: Got {cbs} conditionings but batch-size is {batch_size}")
            else:
                if conditioning.shape[0] != batch_size:
                    print(f"Warning: Got {conditioning.shape[0]} conditionings but batch-size is {batch_size}")


It already gives an error here because the sampler expects either a tensor or a dictionary.

I fixed these flaws but I do not get good results as shown in the paper. It would be very helpful for reproducibility if the authors can share the working version of their codes.

---

> ### Public Comment · ~Shunsuke_Kitada1 · 2022-11-08
> **About the `skip` argument in the sampler**
>
> Thank you very much for making the implementation available to the public.
>
> In relation to the above comment, I would like to ask the authors about the sampler. There are some parts missing in the published implementation. For example, a new argument, `skip`, has been added to the sampler (`txt2img.py`, `line 600`), but the details of this argument are not clear because the implementation of the sampler is not publicly available.
>
> If possible, we would appreciate it if you could provide us with a reproducible procedure for this skip argument. Thank you for the great work.

---

> > ### Author Response · Authors · 2022-11-15
> > **Thank you for your interest**
> >
> > Hi Mehmet and Shunsuke,
> >
> > Thank you for your interest in our work and our code implementation. We appreciate your efforts in reproducing our code in the supplementary materials. Please note that we only uploaded the important files to demonstrate the core implementation of our method.
> >
> > At this time, we have uploaded a temporary demo code for your reference (https://anonymous.4open.science/r/demo_code-BA4F
> > ). The codebase should work by simply copying these files into the official Stable Diffusion codebase. Please do not release the code, as it is not our final official implementation.  Please be assured that we have a timeline to provide the full codebase with an interactive interface after the anonymity period.
> >
> > Thank you,
> > Paper6016 Authors

---

### Author Response · Authors · 2022-11-15
**General Response**

## Revision Summary

We thank all reviewers for their constructive feedback and comments. We have revised our draft to address reviewers’ concerns. We summarize the revision in the following points.


1. More experiments:
    - As suggested by Reviewer 5ZP1, we present Table 2 to quantify compositionally in concept conjunction using human annotations and a phrase grounding model GLIP as an automatic evaluation metric. Our method improves by 3.5\% in generating complete objects with correct colors.
    - As suggested by Reviewer 5ZP1 and QzN2, we added more automatic evaluation in CC-500 and MSCOCO (for ablation). Unfortunately, we found existing models like CLIP and GLIP are not robust or reliable for compositional evaluation. This conclusion is drawn from the “Human-GLIP consistency rate” in Table 2 and previous work (Thrush et al., 2022).
   - As suggested by Reviewer mgR1 and 5ZP1, we added visualization of attention maps throughout the diffusion process and detailed analysis in Appendix C. The visualization justifies our assumption right below Eq. 4: For the same prompt with the same initialization, the attention maps when using a single value (from the whole prompt, Stable Diffusion) are similar to the maps when using multiple values (from multiple re-aligned text spans, our method). It also supports that layout depends on keys while values provide image content.
    - As suggested by Reviewer QzN2 and 5ZP1, we added scene graph setting evaluation in Sec. 3.5 with a comparison to constituency parsing in Appendix D.2. We found that the scene graph setting marginally outperforms Stable Diffusion, potentially due to excessive text spans and imbalance between noun phrases.
    - As suggested by Reviewer QzN2, 5ZP1, and ZtBV, we supplemented more qualitative results in Fig. 12-15 with prompts from ABC-6K or prompts that characterize a wide range of attributes beyond colors, including shape, size, and materials.
    - As suggested by Reviewer ZtBV, we supplemented qualitative results in Fig. 16 to show that our method works with appended style descriptions. In real applications, we may ask users to explicitly indicate the style spans so that they will not be used to create extra values.
    - We added Sec. 4.1 to prove the importance of sequence alignment in maintaining image quality. Without sequence alignment, the system-level image quality drops by ~2 IS/FID.


2. Clarity:
    - As pointed out by Reviewer ZtBV and mgR1, we corrected typos (column headers) in Table 1.
    - As suggested by Reviewer 5ZP1 and ZtBV, we simplified notations and equations in Sec. 2 and revised Fig. 2 & 3 and Algo. 1 for consistency.
    - We have addressed most of the reviewers’ comments regarding clarity and presentation in all sections. We marked the revised part in red.


## Response to the public

We have responded to questions from the public regarding code reproducibility and provided a runnable preliminary codebase. We thank the public for their interest in our work. Please be assured that we do have a timeline to publish our code after the anonymity period.

## General Response

In addition, we would like to respond to some concerns regarding our contribution. We respectfully disagree that our contribution is limited. We are the first to propose a training-free method to address attribute binding and image compositionality. We propose to utilize structured representation from the language domain and blend it into the diffusion guidance process to enhance object-level semantics. Our method effectively improves compositionality from multiple aspects and discovers the importance of individual token embeddings for semantic guidance. Together with extensive analysis, our work is crucial for a deeper understanding of text encoders and cross-attention in diffusion guidance. Our work may also provoke interest in multiple directions, including solving attribute binding issues in a data-efficient manner and robust automatic evaluation for compositionally.

## Nov. 18 update

As the discussion deadline is approaching, **we kindly invite all reviewers to check out our newer version of the paper.** It includes more quantitative and qualitative results and visualization analysis to prove the effectiveness of our method.

---

### Decision · Program_Chairs · 2023-01-20

**Decision:**

Accept: poster

**Justification For Why Not Higher Score:**

The proposed method is novel and interesting, but gives fairly small gains over the baseline.

**Justification For Why Not Lower Score:**

The paper addresses an important issue with text-to-image diffusion models, proposes an easy idea to improve it, and performs extensive experiments to validate their approach. The paper addresses an interesting problem, and has no glaring flaws or errors.

**Metareview: Summary, Strengths And Weaknesses:**

The paper introduces a method for improving attribute binding in text-to-image diffusion models, which is applied to improve the public Stable Diffusion model. The paper also introduces two benchmark datasets to evaluate their method.

Overall all reviewers appreciated the novelty of the problem statement, the fact that the method requires no new training data, and the introduction of the benchmark datasets used for evaluation. There were several issues raised by reviewers that were mostly addressed by revisions to the paper during the rebuttal period, including results on attributes other than colors, results with scene graphs, and automatic evaluation via GLIP. The primary remaining issues were issues with the user study (raised by Reviewer ZtBV) and the sentiment that the proposed method only offered relatively small improvements over the baseline model. Reviewer 5ZP1 was also concerned that the authors had made too many edits to the paper during the rebuttal period.

The AC met with Reviewers 5ZP1 and ZtBV to discuss this paper in more depth. During the discussion, we concluded that the revisions made by the authors during the rebuttal period were not a concern, and indeed had strengthened the paper. Both reviewers praised the overall idea behind the paper and the fact that the proposed method does not require any additional training data. The main weakness of the paper remained its relatively marginal improvements over the baseline.

In addition, Reviewer ZtBV raised an important concern about the user study in Table 1. The paper states that the authors “filter out 20% of the most similar pairs [of images]” before performing the user study. First, this omits the important detail of how “similar image pairs” was identified. Second, under the assumption that similar images would have likely resulted in ties (as the authors argue in their response), this exaggerates the difference between the two methods in Table 1. This could have been remedied either by (a) running the user study on all image pairs, even similar ones; or (b) add an extra column to Table 1 (Win / Lose / Tie / Similar Image) to give a more accurate sense of the relative performance of the models.

Overall, the AC feels that the paper’s novelty outweighs its relatively small improvements over the baseline method. The proposed method is easy to adapt to other models, and more importantly contributes an interesting set of new ideas that may spark interesting conversations and follow up work in the community. Reviewers 5ZP1 and ZtBV agreed with this viewpoint, and agreed that accepting the paper will benefit ICLR.

The authors are strongly encouraged to take the feedback from reviewers into account when preparing the camera-ready version of the paper, especially the issues with the user study in Table 1 discussed above.


**Note From Pc:**

if the above contains the word "oral" or "spotlight" please see: "oral" presentation means -> notable-top-5% and "spotlight" means -> notable-top-25%. As stated in our emails, we are disassociating presentation type from AC recommendations

**Summary Of Ac-Reviewer Meeting:**

The AC met with Reviewers 5ZP1 and ZtBV. We discussed the role of paper edits during the review process and concluded that the additions to this paper were not a concern. We also discussed how to weigh the paper's novelty against it's small improvements over Stable Diffusion, and concluded that the novelty outweighed the marginal experimental gains. We also discussed issues with the user study in Table 1, and concluded that this was mostly an issue of better presentation and not a core methodological flaw.